# Citrullination modulates antigen processing and presentation by revealing cryptic epitopes in rheumatoid arthritis

Ashley M. Curran [1], Alexander A. Girgis [1,2], Yura Jang[3,6],
Jonathan D. Crawford [1], Mekha A. Thomas [1], Ryan Kawalerski[2,4],
Jeff Coller [4,5], Clifton O. Bingham III[1], Chan Hyun Na [3] & Erika Darrah [1] ✉

Cryptic peptides, hidden from the immune system under physiologic conditions, are revealed by changes to MHC class II processing and hypothesized to drive the loss of immune tolerance to self-antigens in autoimmunity. Rheumatoid arthritis (RA) is an autoimmune disease characterized by immune responses to citrullinated self-antigens, in which arginine residues are converted to citrullines. Here, we investigate the hypothesis that citrullination exposes cryptic peptides by modifying protein structure and proteolytic cleavage. We show that citrullination alters processing and presentation of autoantigens, resulting in the generation of a unique citrullination-dependent repertoire composed primarily of native sequences. This repertoire stimulates T cells from RA patients with anti-citrullinated protein antibodies more robustly than controls. The generation of this unique repertoire is achieved through altered protease cleavage and protein destabilization, rather than direct presentation of citrulline-containing epitopes, suggesting a novel paradigm for the role of protein citrullination in the breach of immune tolerance in RA.

Major histocompatibility complex (MHC) class II antigen processing is a complex system heavily impacted by the structure of proteins entering the pathway[1,2]. Studies have shown that even minor alterations of proteins, such as those introduced by post-translational modifications (PTMs), can have profound effects on protein structure and change the susceptibility of residues to critical proteases[3–8]. Collectively, these changes can dramatically impact downstream antigen processing, leading to the generation of unique T-cell epitopes[3–5]. Modification of self-proteins is hypothesized to lead to both the reduced presentation of dominant self-peptides used to mediate T-cell tolerance and the enhanced presentation of previously cryptic self-peptides[9]. Cryptic peptides are defined as those that are processed and presented from a native antigen inefficiently and at insufficient levels to induce central tolerance under physiologic conditions[9–11]. Autoreactive T cells specific for cryptic epitopes are believed to initiate autoimmunity under conditions that alter the processing and presentation of the native protein[9,12]. Thus far, a comprehensive study of PTM-modulated antigen processing investigating the mechanisms underlying the generation of autoantigen-derived peptide repertoires and the exposure of immunogenic cryptic epitopes has not been undertaken.

Rheumatoid arthritis (RA) is a systemic autoimmune disease characterized by immune-mediated damage to synovial joints, in which immune responses to modified self-proteins are a hallmark finding[13]. In RA, autoantibodies are generated against a group of

[1]Rheumatology, Johns Hopkins University School of Medicine, Baltimore, MD, USA. [2]Biomedical Engineering, Johns Hopkins University, Baltimore, MD, USA. [3]Neurology, Institute for Cell Engineering, Johns Hopkins University School of Medicine, Baltimore, MD, USA. [4]Molecular Biology and Genetics, Johns Hopkins University School of Medicine, Baltimore, MD, USA. [5]Biology, Johns Hopkins University, Baltimore, MD, USA. [6]Present address: Laboratory of Immunology, Office of Biotechnology Products, Center for Drugs Evaluation and Research, Food and Drug Administration, Silver Spring, MD, USA. ✉e-mail: edarrah1@jhmi.edu

proteins in which positively charged arginine residues have been post-translationally converted to the neutral, non-classical amino acid citrulline in an enzymatic process known as citrullination[14,15]. Citrullination is mediated by the peptidylarginine deiminase (PAD) family of enzymes, with the PAD2 and PAD4 isoenzymes most strongly implicated in RA pathogenesis[16]. Although anti-citrullinated protein antibodies (ACPAs) occur in more than 70% of RA patients, the early, initiating events that result in the development of this autoimmune response are still poorly understood. ACPA responses in RA are strongly correlated with a group of MHC class II variants referred to as shared epitope (SE) alleles, thus named for the presence of a shared motif in their peptide-binding grooves[17,18]. Although this strong immunogenetic association, together with the class-switched and high-affinity nature of ACPAs[19], implicates HLA-DR presentation of peptides derived from citrullinated autoantigens in RA pathogenesis, the effect of citrullination on antigen processing has not been studied.

Here, we investigate the hypothesis that citrullination exposes cryptic epitopes, which is sufficient to activate a previously ignorant repertoire of self-reactive T cells and initiate autoimmunity in patients with RA. To achieve this, we use complementary in vitro proteolytic mapping, in silico structure prediction, and natural antigen processing approaches to elucidate the underlying molecular mechanisms of citrullination-modulated antigen processing. We show that citrullination globally modifies antigen processing and presentation through alterations to the structure and proteolysis of citrullinated antigens, promoting the generation and presentation of novel cryptic epitopes capable of stimulating autoreactive T cells from patients with RA. Importantly, the citrullination-induced cryptic peptides are primarily comprised of native (i.e., unmodified) sequences, suggesting a novel paradigm for the role of protein citrullination in the breach of immune tolerance in RA.

## Results

### Citrullination alters antigen processing, resulting in the simultaneous generation of cryptic peptides and the destruction of previously dominant peptides

To investigate the effect of citrullination on antigen processing, we designed an in vitro proteolytic mapping (ProtMap) assay, which utilizes three lysosomal cathepsins (B, S, and H; BSH). This cocktail of cathepsins has been shown to recapitulate the essential features of MHC class II processing[20] and is sufficient to generate immunodominant peptides from autoantigens[21]. We digested native and citrullinated versions of three well-characterized RA autoantigens, which are targeted by autoantibodies in their citrullinated forms—fibrinogen (a trimer of three polypeptide chains: α, β, and γ), vimentin, and hnRNP A2/B1[22–24]. Citrullinated forms of each autoantigen were generated in vitro using purified recombinant human PAD2 or PAD4, the two PAD enzymes associated with RA pathogenesis[24–26]. Citrullination of each antigen was confirmed by western blotting (Supplementary Fig. 1a), and citrullination sites were mapped by mass spectrometry (Supplementary Data 1 and Supplementary Fig. 2). All antigens contained citrullination sites, except for the γ chain of fibrinogen. Following antigen processing with the BSH cathepsin cocktail, proteolysis was confirmed by SDS-PAGE (Supplementary Fig. 1b), and the resulting peptide repertoires were sequenced by label-free quantitation–mass spectrometry (LFQ–MS; Supplementary Data 1). Under conditions that alter antigen processing, we would expect changes to the number of individual peptides or the patterns of peptide generation across the linear protein sequence.

We observed marked qualitative and quantitative changes in the peptide repertoires generated following citrullination for all three RA autoantigens (Fig. 1a and Supplementary Fig. 3a). To visualize changes in processing, the processed peptide repertoires were mapped to their location on the linear protein sequence, and the abundance of all peptides spanning each amino acid residue was calculated for each antigen. This analysis revealed widespread changes to the peptide repertoire across the protein sequence upon citrullination (Fig. 1a). We then defined two potential peptide fates within which we could frame changes in the generation of peptides in response to citrullination throughout the study: (1) creation, which refers to the unique or elevated generation of peptides in the citrullinated samples, and (2) destruction, which refers to the loss or decreased generation of peptides in citrullinated samples. Both the creation and the destruction of various regions within the linear protein sequence were observed, with 9–36% of the sequence of the citrullinated antigens—vimentin, fibrinogen α and β, and hnRNP A2/B1—being significantly changed in abundance by citrullination, despite only 0.8–6.4% of the sequence being directly post-translationally modified by citrullination (Fig. 1a, Supplementary Fig. 3a, and Supplementary Table 1). This phenomenon was also observed for individual peptides detected by mass spectrometry, with 7–49% of peptides being significantly enhanced or created by citrullination and 5–18% being significantly decreased or destroyed, depending on the antigen (Fig. 1b, Supplementary Fig. 3b, and Supplementary Table 1). Interestingly, although fibrinogen γ was not directly citrullinated, 5–26% of its sequence was significantly changed in abundance, which suggests that citrullination of the other fibrinogen chains induced sufficient intermolecular alterations to impact cleavage of the γ chain. These results demonstrate that citrullination induces global changes in the peptide repertoire generated by processing with MHC class II-associated proteases, resulting in both the creation and destruction of numerous peptides derived from diverse regions across the protein sequence landscape.

### Citrullination induces both proximal and distal changes in processing relative to citrullinated sites in the linear amino acid sequence

To investigate the relationship between citrulline residues and the observed changes in antigen processing using our ProtMap data, we first quantified the proportion of significantly changed regions that contained a citrulline residue. Changed regions were defined as contiguous stretches of amino acid residues in the linear sequence with at least twofold enrichment or depletion of peptide abundance in the PAD2- or PAD4-citrullinated sample compared to the corresponding native sample. Only 21–31% of created regions and 24–37% of destroyed regions contained citrulline residues, indicating that citrullination impacts antigen processing more broadly than through the mere generation of novel citrulline-containing peptides (Fig. 2a). There are then two, non-mutually exclusive mechanisms through which citrullination could potentially impact antigen processing: citrulline residues outside but directly proximal to changed regions affecting proteolytic cleavage, or distal citrulline residues mediating structural changes.

To examine whether citrulline residues were associated with proximal changes, we compared the frequency of citrullines immediately adjacent to significantly changed regions in the linear sequence to the expected frequency of citrulline residues within the same size residue window for all antigens. We found that citrullines were significantly enriched compared to the expected frequency within a four-residue window surrounding newly created cut sites (43.0 vs. 29.7, $P = 0.001$) but not destroyed cut sites (25.0 vs. 26.7, $P = 0.678$). In newly created cut sites containing citrulline residues, citrullines were significantly enriched at the P1 and P2' positions ($P = 0.05$ and $P = 0.003$, respectively; Fig. 2b). Therefore, citrullines are disproportionately enriched within novel cleavage sites in citrullinated antigens, suggesting that citrulline residues have a proximal impact on processing by MHC class II-associated proteases. However, as many of the citrullination sites existed outside of differentially abundant or changed regions (Fig. 1a–c), we next investigated the relationship between changed regions and all citrulline residues in the linear amino acid sequence. We observed

both citrulline-proximal (i.e., actual minus expected distance < 0) and citrulline-distal (i.e., actual minus expected distance > 0) changed regions, which occurred in a non-normal distribution around the expected distance (Shapiro–Wilk test, $P < 0.0001$; Fig. 2c). Thus, protein citrullination results in the creation and destruction of peptides around citrulline residues, as well as in areas that are distant from citrulline residues in the linear protein sequence.

### Citrullination impacts protein dynamics by promoting the destabilization of protein folding

The observation that distal changes in antigen processing occur relative to citrulline residues suggests that citrullination promotes structural changes responsible for the exposure or obstruction of peptides derived from distal protein regions. Interestingly, the citrullinated autoantigens used in this study had higher than average arginine content for vertebrate proteins, representing elevated citrullination potential, which we hypothesized might promote

protein unfolding and autoantigenesis (Fig. 2d)[27,28]. We found a weak positive correlation between arginine frequency and the extent of changes to antigen processing, which we represented as an antigen processing change score defined as the proportion of all identified peptides that were significantly changed in abundance for each antigen. However, only 44–60% of the variance in antigen processing change scores could be explained by arginine frequency alone (Fig. 2e), implicating more nuanced structural changes in addition to citrullination potential.

To investigate whether antigen processing changes could be explained by citrullination-induced structural reorganization, we utilized AlphaFold v2.0 to model both native and citrullinated protein structures in silico[29,30]. To date, the structures of citrullinated proteins have not been determined experimentally, and several of the proteins used in this study have not been fully crystallized even in their native forms. Given these limitations, AlphaFold predictions allowed for the direct comparison of native and citrullinated protein structures within

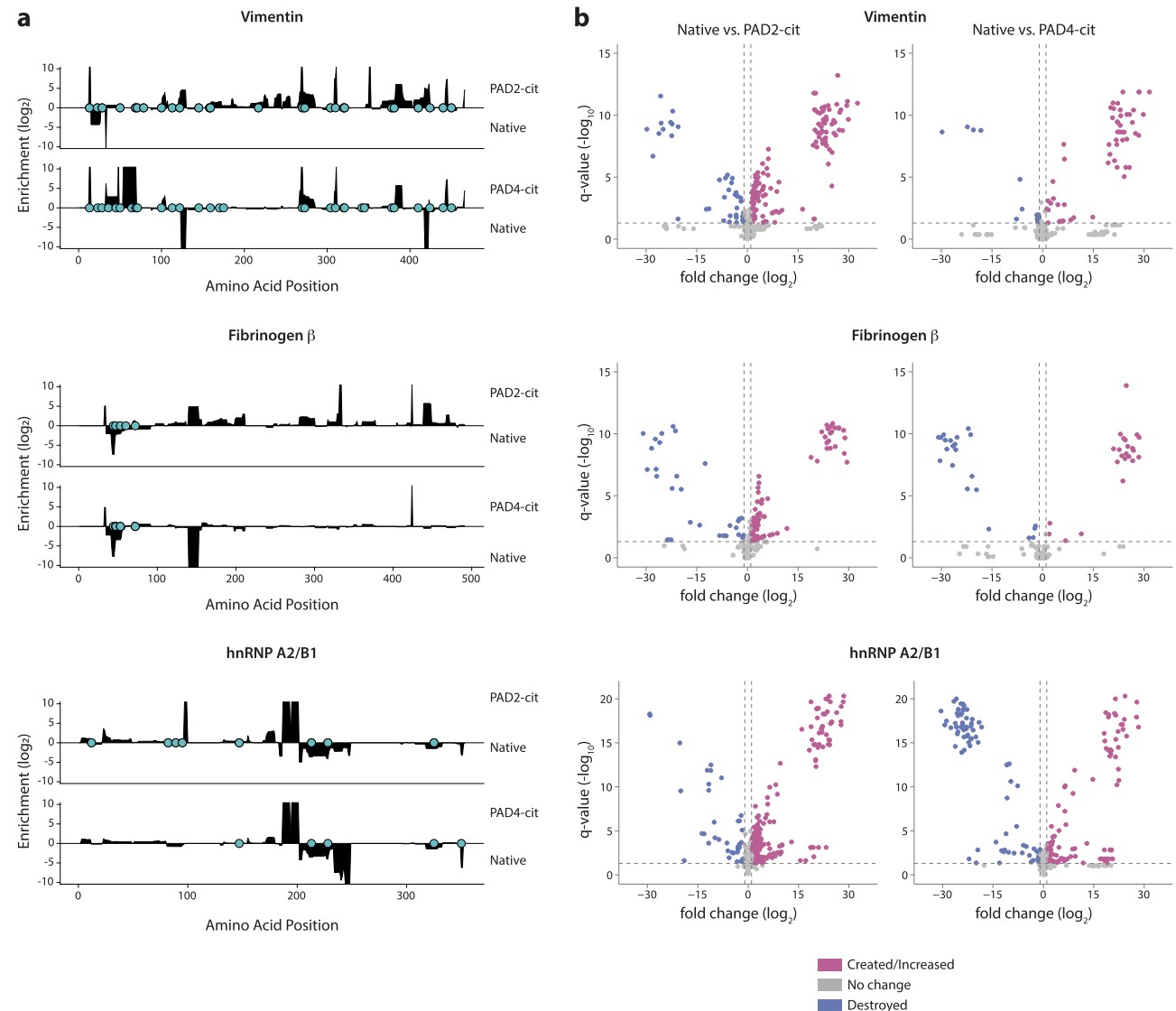

**Fig. 1 | Citrullination alters antigen processing, resulting in the simultaneous generation of cryptic peptides and the destruction of previously dominant peptides. a** Log₂ enrichment of peptides from proteolytic mapping with cathepsins BSH across the primary amino acid sequence of each protein. Amino acid positions with positive values are enriched in the PAD2- or PAD4-citrullinated (cit) samples compared to the native (nat) samples, while those with negative values are reduced by citrullination. Citrulline residues are denoted by the filled blue circles on x axes. **b** Volcano plots representing peptides with significantly altered abundance. Vertical lines denote a fold change of 2, and the horizontal line denotes a cutoff of 0.05 for the FDR-corrected $P$ value ($q$ value), calculated by paired two-sided Student's $t$ tests. Peptides to the left or right of vertical dashed lines and above horizontal dashed line are deemed to be significantly different between the groups. Proteolytic mapping was performed in replicates of four (fibrinogen and vimentin) or six (hnRNP A2/B1).

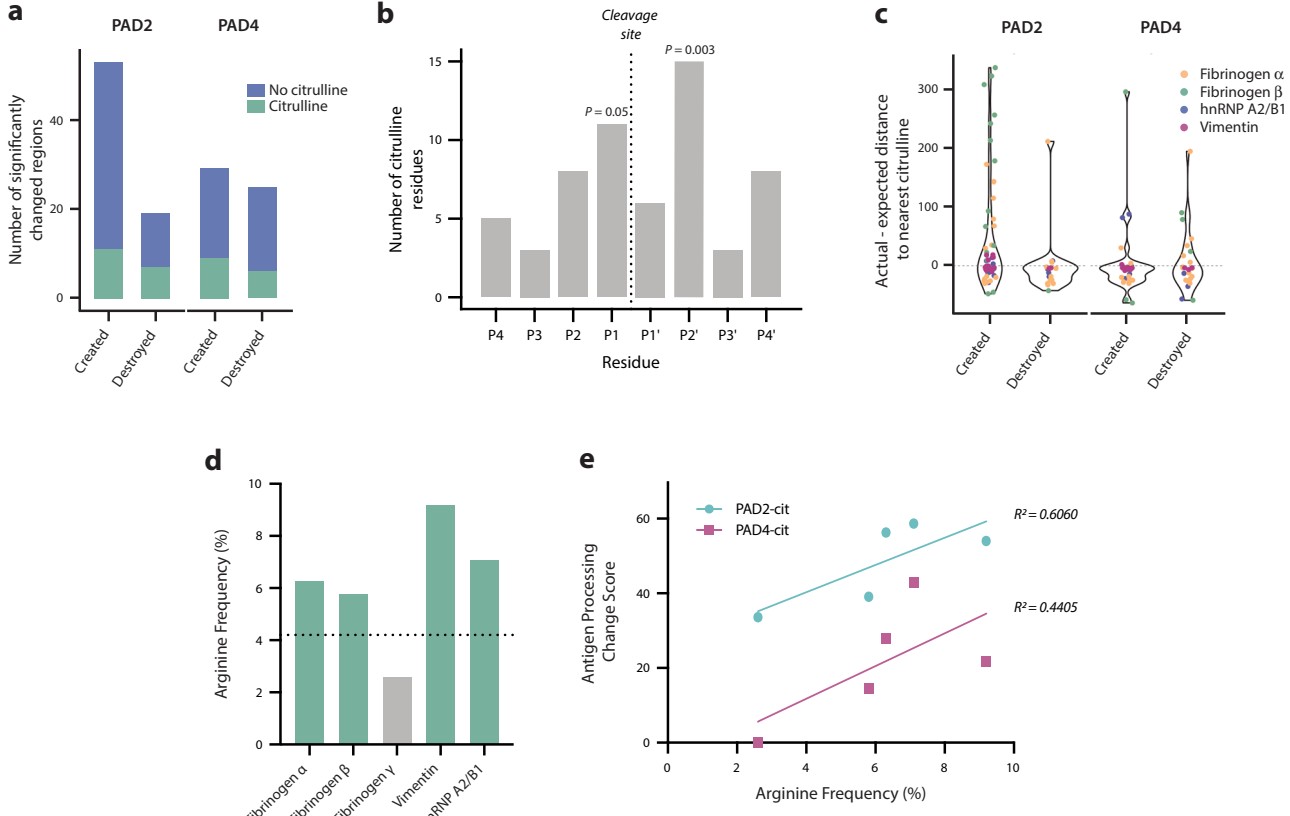

**Fig. 2 | Citrullination induces proximal and distal changes in processing relative to citrullinated sites. a** Percentage of significantly created and destroyed regions derived from PAD2- and PAD4-citrullinated antigens that contain citrulline residues. **b** Frequency of citrulline residues in each position (P4-P4') of citrulline-containing cleavage sites enriched in the citrullinated antigens. *P* values were calculated via bootstrapping to compare the observed frequency of citrullines to the expected frequency of citrullines in each position. Only significant *P* values (≤0.05)

are shown. **c** Violin plots depicting the actual minus expected distance from each significantly changed region to the nearest citrulline residue in the linear amino acid sequence. **d** Arginine frequency shown for all autoantigens used in this study. The horizontal dotted line represents the average vertebrate protein arginine content. **e** Antigen processing change score versus % arginine content (frequency) for each PAD2- or PAD4-citrullinated (cit) antigen.

the same highly accurate modeling system. Structural alignment revealed visible differences between the native and citrullinated protein structure predictions (Fig. 3a). We calculated the root-mean-square deviation (RMSD)[31] and template modeling score (TM-score)[32,33] to quantify deviations between the modeled native and citrullinated structures for each autoantigen (Supplementary Table 2). We measured RMSD values of 6–27 Å when comparing the predicted structures of the native and citrullinated antigens, far exceeding the generally accepted threshold of <2 Å for similar folds (where RMSD = 0 indicates perfect similarity). Although RMSD is the most common measure of structural similarity, this calculation can be driven by large changes in small unstructured domains as the deviation of each atom is equally weighted[34]. Similarly to experimental methods commonly used to determine protein structure, AlphaFold predicts the conformation of unstructured or intrinsically disordered protein domains with low confidence; thus, predicted misalignment between native and citrullinated proteins in these domains are more likely to be driven by stochastic differences in conformational predictions. To validate our findings, we additionally calculated the TM-score, which weights smaller deviations more heavily than larger ones, which are more likely to be found in unstructured domains, and better controls for protein length to enable interprotein comparison. In this method, TM-scores can range from 0-1, where 1 indicates perfect similarity between two structures and <0.5 represents highly dissimilar proteins. For our citrullinated antigens, we calculated TM-scores of 0.6–0.89 (Supplementary Table 2). Considering how few residues differ between our otherwise identical native and citrullinated sequences, TM-scores as

low as 0.6 demonstrate the potential for very few post-translationally modified sites to induce a disproportionately large change in structure.

Visualization of cathepsin BSH-created and -destroyed regions of each protein derived from ProtMap on the predicted citrullinated protein structure revealed the destruction of peptides derived from unstructured domains and the creation of peptides located within regions of secondary structure (e.g., alpha-helices) and globular domains (Fig. 3b and Supplementary Fig. 4). To investigate the correlation between changed regions and citrulline residues in the 3-dimensional (3D) protein structure, we calculated the Euclidean distance between each changed region and the nearest citrulline residue, relative to the expected distance. Similar to the pattern observed for linear distances of changed regions to citrullines (Fig. 2c), we found both citrulline-proximal and citrulline-distal changed regions and observed different distributions between created and destroyed regions, with created regions more frequently distal to citrullines and destroyed regions more frequently proximal (Fig. 3c). This finding is consistent with the higher frequency of citrullines within destroyed regions (Fig. 2a). Furthermore, the persistence of changed regions distal to citrulline residues in the 3D analysis emphasizes the role of citrullination in inducing structural changes that can impact antigen processing from afar.

To evaluate the contribution of these predicted, citrullination-induced structural changes to the extent of alterations in antigen processing, we measured the correlation between the RMSD and TM-score values and the corresponding antigen processing change scores for

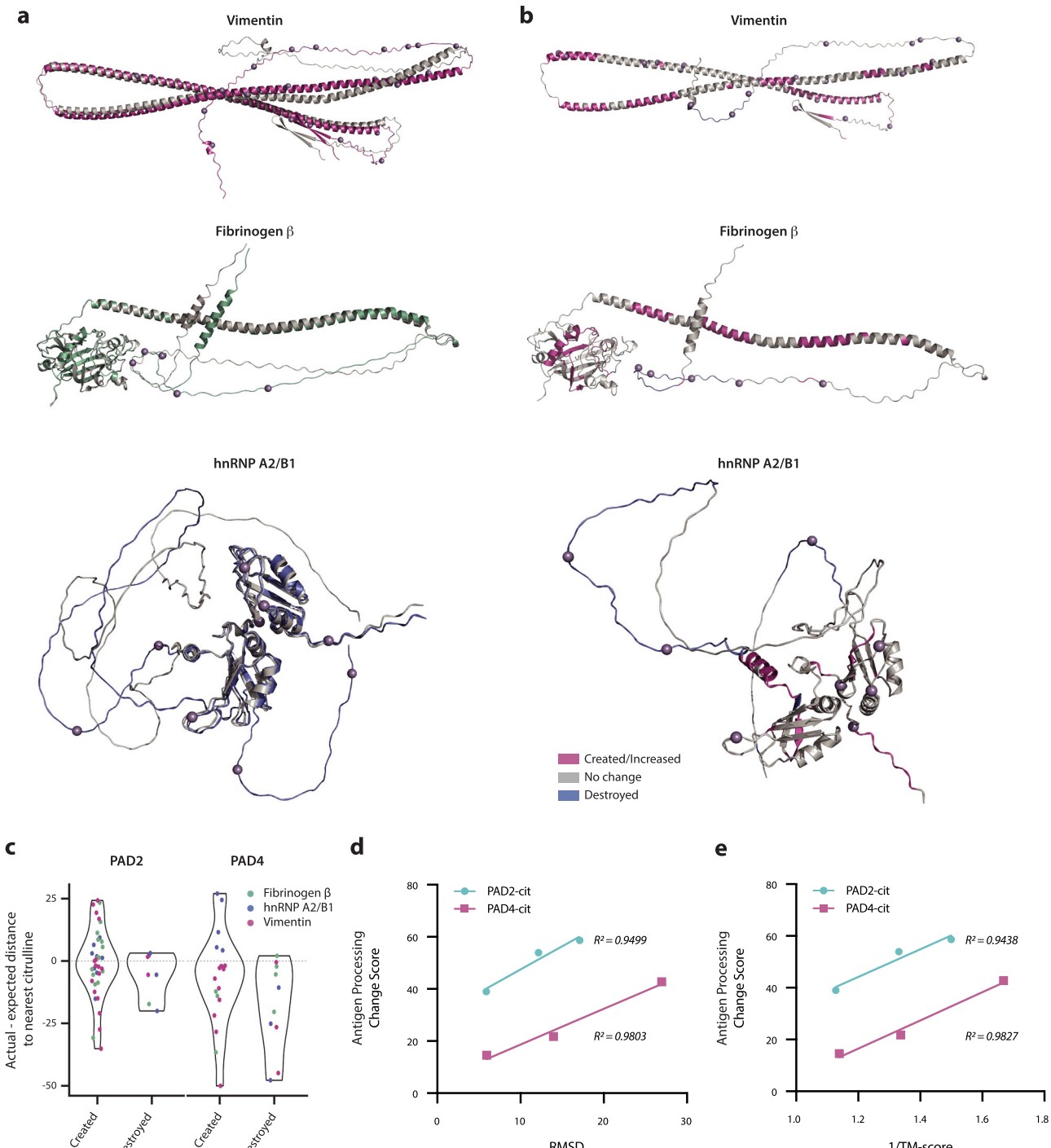

**Fig. 3 | Citrullination impacts protein dynamics by promoting the destabilization of protein folding. a** PyMOL structural alignment of native (gray) and PAD2/4-combined citrullinated (colored by antigen) PDB structures predicted by AlphaFold AI system, excluding fibrinogen α chain due to computational constraints. Mass spectrometry-confirmed citrullination sites from PAD2 and PAD4 combined are denoted as purple spheres on the citrullinated structure. **b** Predicted PAD2-citrullinated structures with significantly changed regions color-coded according to each antigen. We observed a positive relationship between both measures of structural deviation and the extent to which antigen processing was impacted by citrullination, wherein up to 98% of the variance can be explained by changes in structure (Fig. 3d, e). Together, these findings strongly suggest that citrullination modulates antigen processing through protein destabilization, in addition to altered susceptibility to proteolytic cleavage.

legend and PAD2 citrullination sites denoted as purple spheres. See Supplementary Fig. 4 for PAD4-citrullinated structures. **c** Violin plots depicting the actual minus expected distance from each significantly changed region to the nearest citrulline residue in the 3-dimensional structure. **d, e** Antigen processing change score versus RMSD value or 1/TM-score, respectively, analyzed by simple linear regression for PAD2- or PAD4-citrullinated (cit) vimentin, fibrinogen β, and hnRNP A2/B1.

## Putative RA-associated HLA-DR peptide-binding cores primarily consist of native sequences enriched by citrullination

To explore the potential impact of citrullination-modulated antigen processing on HLA-DR selection, we elucidated the predicted contributions of citrulline-containing versus native epitopes to a unique HLA-DR–presented repertoire. Peptides identified by proteolytic mapping were evaluated by the NetMHCII-2.3 binding prediction

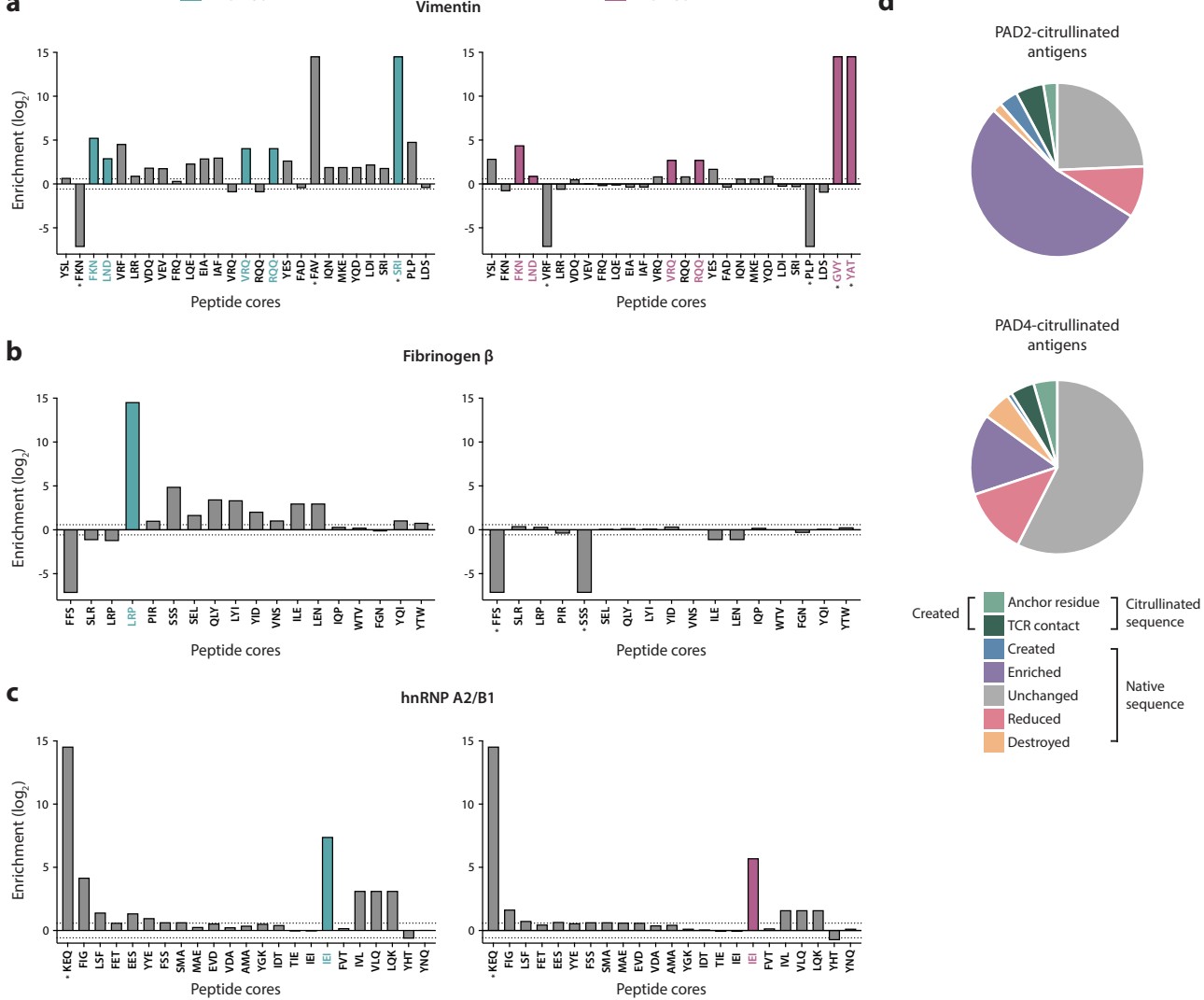

**Fig. 4 | Putative RA-associated HLA-DR peptide-binding cores primarily consist of native sequences enriched by citrullination. a–c** Log₂ enrichment of putative binding cores (<500 nM) predicted by the NetMHCII-2.3 algorithm, denoted by the first three amino acids of the peptide core. Cores with positive values are enriched in the PAD2- or PAD4-citrullinated (cit) samples, while those with negative values are reduced by citrullination. *Denotes a uniquely created or destroyed peptide, and color-coded cores are those that contained a citrullination site. **d** Pie charts representing the proportion of all putative binding cores belonging to several categories (given by legend) denoting the behavior of each core in response to citrullination.

algorithm[35], which is based on empirically-derived peptide-MHC binding affinities from the Immune Epitope Database (IEDB)[36]. We predicted the binding affinity to all available RA-associated SE alleles (i.e., DRB1*01:01, *04:01, *04:04, *04:05, and *10:01) for the peptides derived from native and citrullinated antigens[37,38]. Peptide-binding cores were extracted from peptides that were considered putative binders (<500 nM binding affinity) to any SE allele, and the abundance of each putative binding core was quantified in the aggregate peptide data. Enrichment of each putative peptide-binding core in the native or citrullinated samples is shown for each antigen (Fig. 4a–c and Supplementary Fig. 5a, b).

Most putative binding cores were enriched in the citrullinated samples and consisted largely of native amino acid sequences not containing citrulline residues (Fig. 4a–c and Supplementary Fig. 5a, b). Among all putative binding cores, 11% (13/115)–18% (20/113) were reduced or destroyed by citrullination, and 25% (28/113)–64% (74/115) were enriched or created by citrullination (Fig. 4d and Supplementary Table 3). Of the enriched or created cores, only 12% (9/74)–36% (10/28) contained citrulline residues, and of those containing citrullines, 33% (3/9) and 50% (5/10) generated by PAD2 and PAD4, respectively, were

predicted to contain citrullines as anchor residues. This phenomenon persisted whether putative binding cores were aggregated for all SE alleles or analyzed independently for each allele (Supplementary Fig. 5c). Consistent with our previous findings, these results demonstrate that most of the changes in antigen processing induced by citrullination lead to the novel or enhanced generation of native, rather than citrulline-containing putative epitopes. The enrichment of putative binding cores in the citrullinated samples indicates that the unique, citrullination-dependent peptide repertoire is likely to bind with higher affinity to RA-associated HLA-DR alleles. This finding is supported by the observation that the median predicted binding affinities of the created peptide repertoires to RA-associated HLA-DR variants is elevated compared to the destroyed peptide repertoires (Supplementary Fig. 6).

**Citrullination alters the naturally presented peptide repertoire of human monocyte-derived dendritic cells**
To investigate whether citrullination alters antigen processing and presentation in a cellular model, we utilized a natural antigen processing assay (NAPA), which harnesses the natural processing machinery

of human antigen-presenting cells (APCs) and provides an unbiased approach to determine the presented peptide repertoire derived from any antigen[39]. We performed NAPA on monocyte-derived dendritic cells (mo-DCs) from a healthy donor with one SE allele (HLA-DRB1*04:01) pulsed with fibrinogen and found striking changes in the total HLA-DR−presented peptide repertoires between the native and citrullinated forms (Fig. 5a and Supplementary Table 4). Seven peptide epitopes were presented from native fibrinogen, with peptides being presented from each of the three polypeptide chains. Upon citrullination, three peptides that were dominantly presented from native fibrinogen were destroyed, and three novel, cryptic peptides were revealed. Importantly, however, none of the peptides identified in the citrullinated samples contained citrulline residues. Although PAD2 and PAD4 can generate overlapping yet distinct patterns of citrullination (Fig. 1 and Supplementary Fig. 2), identical peptide repertoires were presented in the two citrullinated samples, indicating a convergent mechanism for citrullination-induced changes in antigen processing and ultimate HLA-DR peptide selection. Further, the presentation of a subset of peptides was enhanced following citrullination, which suggests an alteration in the accessibility of these peptides to MHC molecules. Remarkably, all destroyed peptides contained a citrullination site, suggesting that citrullination resulted in enhanced proteolytic cleavage or decreased MHC binding affinity (Fig. 5a). Together, these findings confirm that citrullination has the capacity to modulate MHC class II antigen processing, through both the creation and destruction of epitopes, leading to widespread alterations in the resulting autoantigen peptide repertoire presented by APCs.

We then measured the relative binding affinity of the NAPA-derived peptides to HLA-DRB1*04:01 using the ProImmune REVEAL® platform to further elucidate the mechanisms responsible for the altered peptide repertoire. In this binding study, the peptide repertoire presented from the citrullinated antigen bound with a higher median relative affinity than the repertoire presented from the native antigen (Fig. 5b and Supplementary Table 5). This in vitro finding agrees with our in silico ProtMap analysis, which predicted that peptide repertoires derived from citrullinated antigens would bind with higher affinity to SE alleles than native antigen-derived repertoires (Supplementary Fig. 6). Since we observed the destruction of all citrulline-containing epitopes, we additionally synthesized and measured the relative binding affinity of the citrullinated versions of the presented native peptides to shed light on the relative contributions of altered processing versus MHC class II binding in their destruction. Interestingly, when we compared the relative binding affinity scores of the three citrulline-containing epitopes and their native counterparts, we observed three distinct outcomes: enhancement, ablation, or no change in binding upon citrullination (Fig. 5c and Supplementary Table 5). This observation is consistent with the current literature, wherein citrulline-containing peptides exhibit higher binding affinities to SE alleles than unmodified peptides in certain biophysical studies but not others[37,40,41]. Thus, the destruction of peptide presentation upon citrullination likely relies on a combination of less efficient MHC class II binding (e.g., peptide 8) and peptide destruction during upstream antigen processing (e.g., peptide 10) preventing stable presentation.

### Unmodified, citrullination-dependent epitopes stimulate ACPA⁺SE⁺ RA patient CD4⁺ T cells more robustly than native antigen−derived epitopes

We reasoned that, if citrullination acts by revealing previously cryptic epitopes to induce loss of tolerance, CD4⁺ T cells from RA patients with anti-citrulline immune responses (i.e., ACPAs) would preferentially respond to the citrullination-dependent peptide repertoire. We synthesized the peptides identified by NAPA (Supplementary Table 7) and stimulated peripheral blood mononuclear cells (PBMCs) from ACPA⁺SE⁺ RA patients ($n = 10$), ACPA⁻SE⁺ RA patients ($n = 8$), and SE⁺ healthy controls ($n = 10$) with peptides from each peptide group (i.e., created/enhanced, no change, and destroyed). Following overnight stimulation, we measured CD154 (CD40L) surface expression by flow cytometry to assess CD4⁺ T-cell activation status. Representative flow plots are shown in Supplementary Fig. 8.

We found that T cells from ACPA⁺ RA patients preferentially recognized created peptides (Fig. 6a). ACPA⁻ RA patients, on the other hand, showed more frequent activation upon stimulation with the unchanged and destroyed peptide groups. The percentage of potential T-cell responses achieved in each category revealed a non-significant trend toward higher responses among ACPA⁺ RA patients to the created peptides (33%) than to no change (17%; $P = 0.2918$) and destroyed (13%; $P = 0.1692$) peptide groups, in addition to significantly more frequent responses among ACPA⁺ RA patients to the created peptides (33%) than observed for the controls by $\chi^2$ test (2.5%; $P = 0.0002$; Fig. 6b). CD154⁺CD4⁺ T cells from ACPA⁺ RA patients exhibited a significant expansion of Th17 and Th1–17 intermediate T-cell helper subsets compared to the total parent CD4⁺ population ($P = 0.0061$ and $P = 0.0365$, respectively), with minimal differences observed between peptide groups (Fig. 6c). The majority of CD154⁺CD4⁺ T cells exhibited a T-helper-cell phenotype, suggesting that they had undergone TCR activation and differentiation in response to antigen stimulation in vivo. Cytokine secretion assays supported these findings, wherein ACPA⁺ RA patient CD4⁺ T cells predominantly secreted Th2- and Th17-type cytokines in response to stimulation with created fibrinogen peptides (Fig. 6d).

### ACPA⁺SE⁺ RA patient CD4⁺ T cells specific for citrullination-dependent epitopes tend to be activated effector memory cells

To further validate our detection of CD4⁺ T cells specific for citrullination-dependent (i.e., created) epitopes, we measured CD4⁺ T-cell binding to MHC class II tetramers containing three of the created fibrinogen peptides (Supplementary Table 8). Representative flow plots are shown in Supplementary Fig. 10. ACPA⁺ RA patients have significantly more frequent tetramer-positive cells compared to ACPA⁻ patients (Fig. 7a, b), again suggesting that these cells are specific to the citrullination-dependent RA disease pathway. We detected similar levels of tetramer-positive cells in ACPA⁺ patients and SE⁺ healthy controls (Fig. 7a, b); however, when we examined the effector phenotypes of T cells in each group, we found that ACPA⁺ patients exhibited a trend toward more central memory T cells ($P = 0.0698$; Fig. 7c), while tetramer-positive cells in SE⁺ healthy controls were significantly more likely to be regulatory T cells ($P = 0.03$; Fig. 7d). In fact, the SE⁺ healthy controls had a mean ± SD of 87.72 ± 13.36% regulatory T cells among fibrinogen tetramer-positive cells compared to 53.92 ± 32.29% in ACPA⁺ patients. This finding is in line with recent studies demonstrating that healthy individuals harbor similar levels of self-reactive and pathogen-specific T cells in their peripheral blood, necessitating the maintenance of self-tolerance by enhanced levels of antigen-specific regulatory T cells[42–45]. Strikingly, we found CD4⁺ T cells specific for just a few native epitopes, identified exclusively from the processing of citrullinated fibrinogen by our natural antigen processing assay, in more than 50% of ACPA⁺ RA patients. Together, these findings support the hypothesis that cryptic peptides revealed by citrullination may play a role in the development of immune responses and loss of immune tolerance to citrullinated antigens in RA.

## Discussion

Despite the appreciation that citrullinated autoantigens are central to RA pathogenesis, the effect of citrullination on antigen processing and presentation, early fundamental events in the initiation of autoreactive immune responses, is unknown. To date, through the study of synthetic peptides identified by prediction algorithms or mouse models, the paradigm holds that recognition of citrulline-containing peptides by autoreactive CD4⁺ T cells is the primary

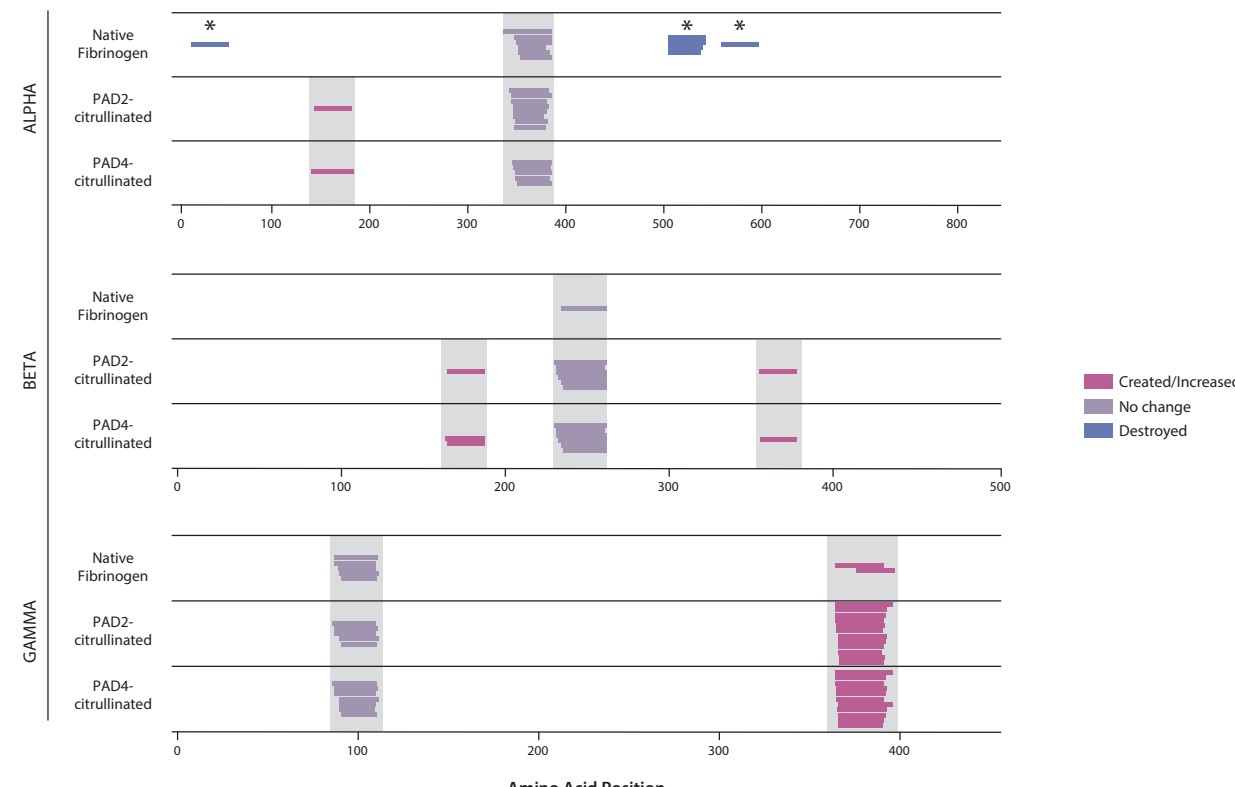

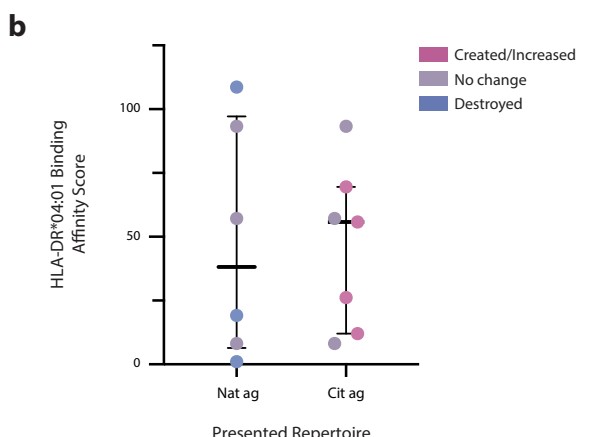

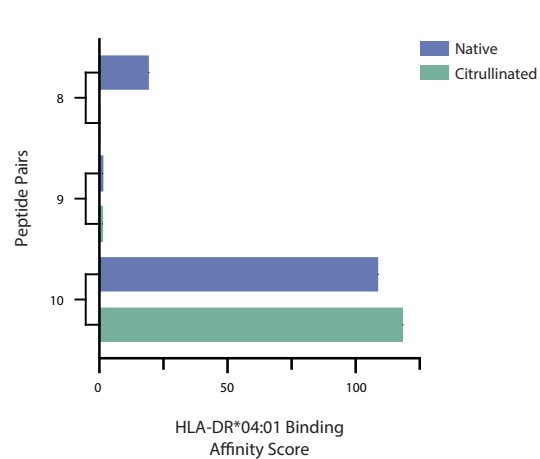

**Fig. 5 | Citrullination alters the naturally presented peptide repertoire presented by human mo-DCs. a** NAPA performed on mo-DCs from a SE[+] healthy donor pulsed with native and citrullinated fibrinogen. Peptides from the α, β, and γ chains of fibrinogen are shown. Each colored line represents a unique peptide coded according to the legend, and gray bars denote peptides with identical core motifs between samples. *Denotes a citrullination site. **b** HLA-DRB1*04:01 relative binding affinity scores for the presented peptide repertoires derived from the native antigen (Nat ag; $n = 6$ peptides) and the citrullinated antigen (Cit ag; $n = 7$ peptides) were measured by the ProImmune REVEAL® assay, which represents a surrogate for binding affinity through the quantification of MHC stabilization relative to an internal positive control peptide. The center lines represent the median, and the error bars denote the interquartile range. **c** HLA-DRB1*04:01 relative binding affinity scores as measured by the ProImmune REVEAL® assay for the three native and citrullinated peptide pairs.

mechanism for the development of anti-citrullinated protein immune responses[37,40,41,46–50]. However, our study challenges the current paradigm that the initiation of autoimmune responses to citrullinated self-antigens in RA relies exclusively on MHC class II presentation of citrulline-containing epitopes. Rather, we show that citrullination has a dramatic impact on antigen processing with effects that far exceed the creation of citrulline-containing epitopes, resulting in the generation of a unique citrullination-dependent peptide repertoire composed primarily of native sequences. Our

findings may thus serve to reconcile the disparate observations that citrulline-containing synthetic peptides exhibit higher binding affinities for SE alleles than unmodified peptides in certain biophysical studies but not others[37,40,41], and that citrullinated peptides have not been isolated from HLA-DR molecules in the RA synovium despite their proposed role in disease initiation and propagation[51,52]. The findings of our study and the current paradigm are not by necessity mutually exclusive, and indeed, our study reconciles the seemingly incongruent observations in the literature by revealing native

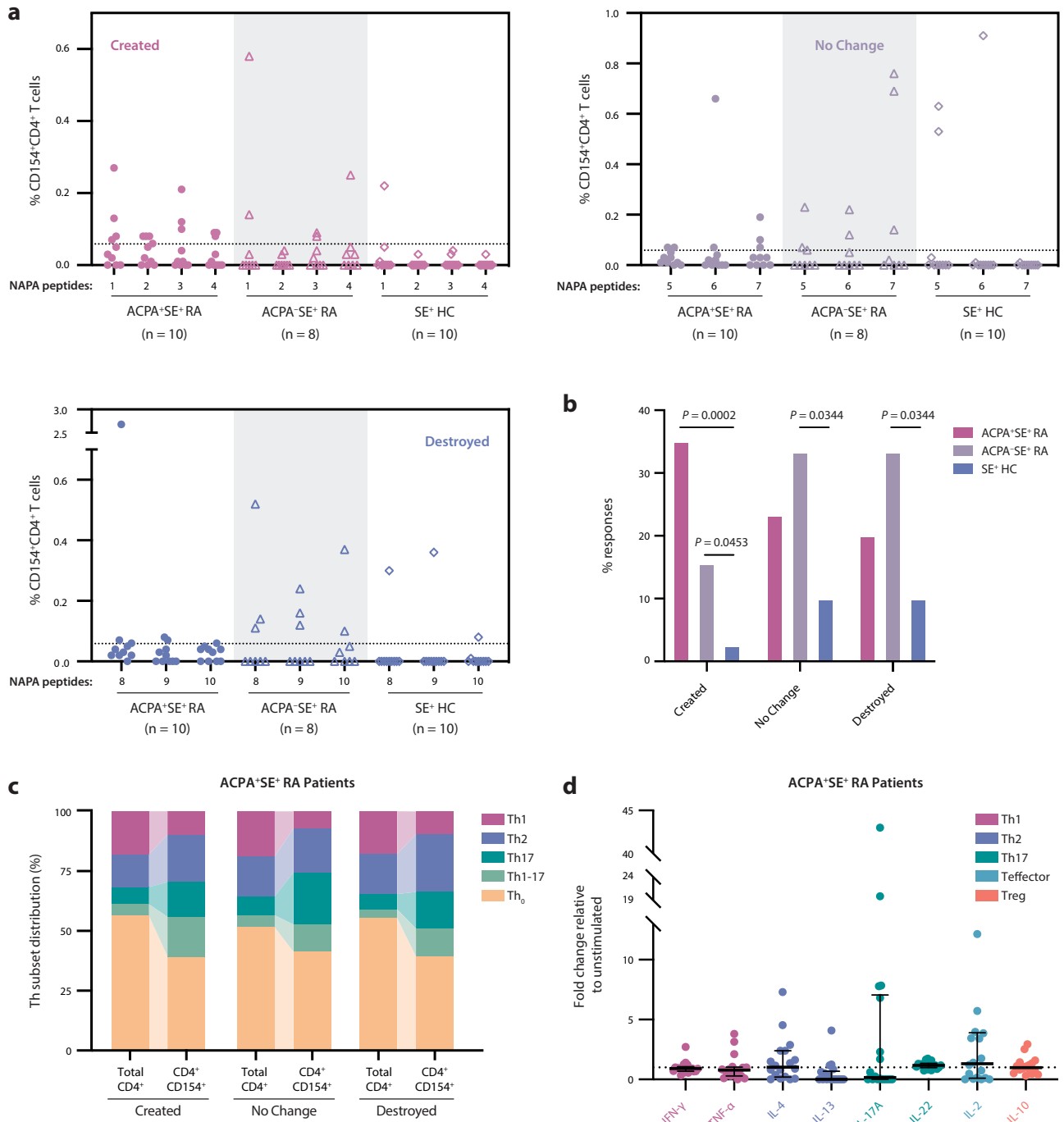

**Fig. 6 | Unmodified, citrullination-dependent epitopes stimulate ACPA⁺SE⁺ RA patient CD4⁺ T cells more robustly than native antigen–derived epitopes.**
**a** PBMCs from ACPA⁺SE⁺ RA patients ($n = 10$), ACPA⁻SE⁺RA patients ($n = 8$), and SE⁺ healthy controls ($n = 10$) were stimulated with created peptides (1–4), no change peptides (5–7), and destroyed peptides (8–10) derived from NAPA (Fig. 5a). Percentage of CD154⁺CD4⁺ T-cell response to each peptide is shown. **b** Percentage of potential T-cell responses (where the number of potential T-cell responses were defined as total patients per group multiplied by the total peptides per group) is shown. Two-sided $\chi^2$ tests were used to compare the frequency of T-cell responses between patient groups. Only significant $P$ values (≤0.05) are shown. **c** Distribution of ACPA⁺SE⁺ RA patient CD4⁺ T-helper subsets (denoted in legend) in either the total CD4⁺ or the CD4⁺CD154⁺ T-cell population in response to stimulation with created, no change, or destroyed peptides. **d** ACPA⁺SE⁺ RA patient ($n = 18$) PBMC cytokine secretion in response to 48-h stimulation with the created peptide pool, normalized to unstimulated samples. The dotted line at a fold change of 1 represents the normalized unstimulated values. Data are presented as median and IQR.

epitopes as potential drivers of autoimmune initiation, perhaps alongside citrulline-containing peptides.

Cryptic peptides, against which immune tolerance does not readily develop, are thought to initiate autoimmunity under conditions that promote altered antigen processing[9,12]. Therefore, any novel peptide presented outside the context of central or peripheral tolerance

mechanisms may appear foreign to the immune system, including novel native sequences uniquely derived from a post-translationally modified protein. Preferential recognition by ACPA⁺SE⁺ RA patient CD4⁺ T cells of the native, citrullination-induced fibrinogen peptide repertoire is evidence that the presentation of these cryptic peptides occurs in these patients. Importantly, this also highlights that the association between

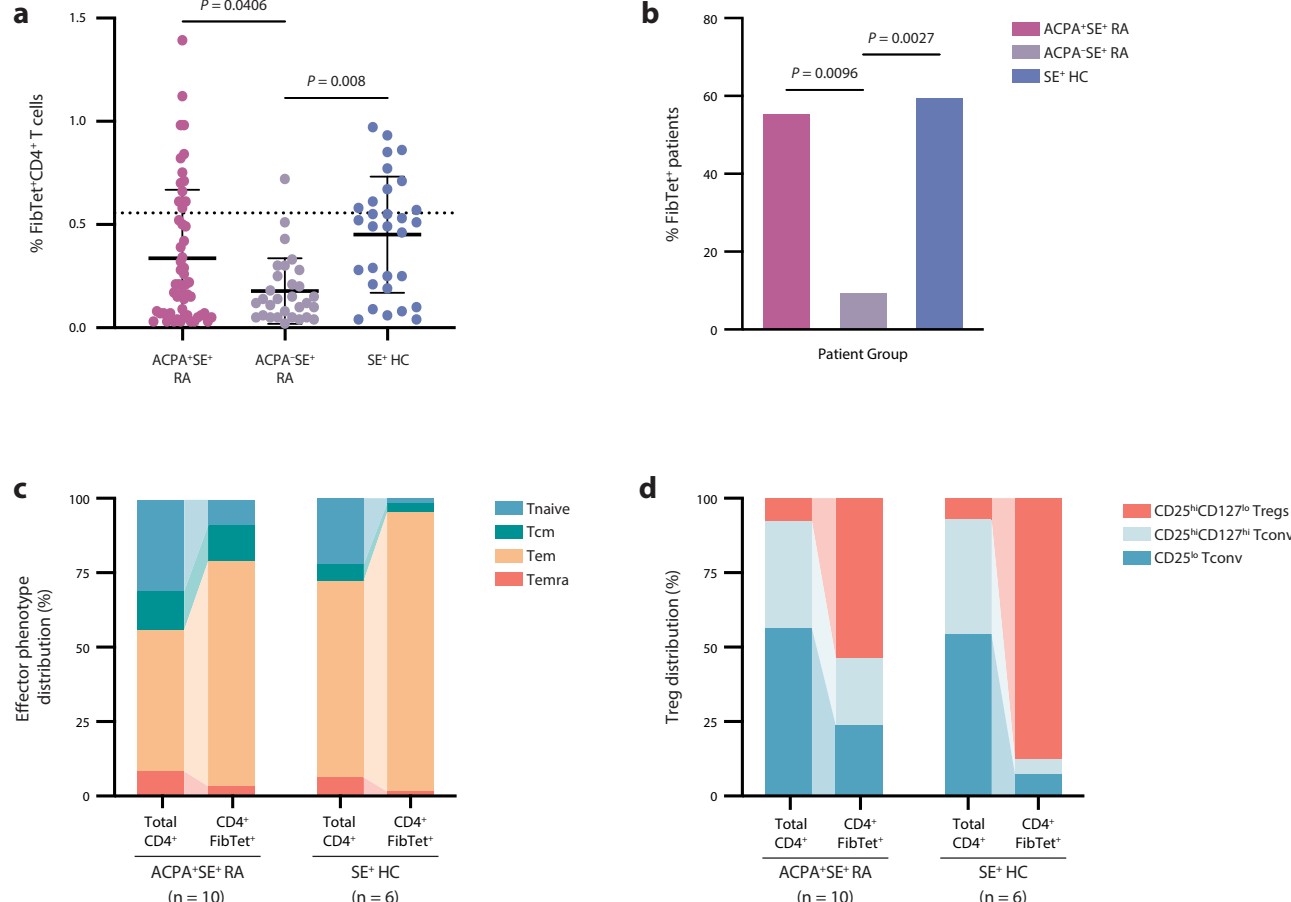

**Fig. 7 | ACPA⁺SE⁺ RA patient CD4⁺ T cells specific for citrullination-dependent epitopes tend to be activated effector memory cells. a** MHC class II tetramers containing three created fibrinogen peptides derived from NAPA (Fig. 5a) were used to stain PBMCs from ACPA⁺SE⁺ RA patients (n = 18), ACPA⁻SE⁺RA patients (n = 10), and SE⁺ healthy controls (n = 10). The percentage of fibrinogen tetramer−positive (FibTet⁺) CD4⁺ T cells are shown for each patient group, and the dotted line denotes the positivity threshold of 0.5547%. Data are presented as mean ± SD, and P values were calculated by ordinary one-way ANOVA with Tukey's multiple comparisons test. Only significant P values (≤0.05) are shown. **b** The percentage of individuals that bound any FibTet is shown for each group. Two-sided $\chi^2$ tests were used to compare the frequency of T-cell responses between patient groups. Only significant P values (≤0.05) are shown. **c, d** Distribution of effector phenotypes (**c**) and regulatory T cells (**d**) for ACPA⁺SE⁺ RA patient and SE⁺ healthy control fibrinogen tetramer−positive T cells. Phenotypes denoted in key.

ACPA⁺ RA and SE alleles does not necessitate enhanced presentation of citrulline-containing peptides by SE alleles. Intriguingly, a recent study demonstrated that autoimmune-associated HLA alleles may preferentially select autoreactive T cells in the thymus during negative selection[53], which could act in conjunction with enhanced binding of cryptic peptides to promote loss of immune tolerance to citrullinated antigens in individuals with predisposing HLA alleles.

We found that citrullination modifies antigen processing by altering susceptibility to proteolytic cleavage in vitro and promoting destabilization of protein folding in silico, leading to both the creation and destruction of regions within self-proteins. Alongside the creation of novel peptides, the destruction of previously dominant epitopes is likely critical to the exposure and selection of cryptic epitopes for MHC class II presentation. Our study revealed that impacts on antigen processing are mediated by a combination of citrulline-proximal and -distal changes, which are influenced by a number of autoantigen-specific factors, such as citrullination potential, protein structure, and accessibility to proteases. Each autoantigen we studied exhibited different biases toward creation versus destruction and proximal versus distal changes. Similarly, the two citrullinating enzymes, PAD2 and PAD4, influenced antigen processing through different mechanisms—PAD4 citrullination induced more frequent proximal changes, while PAD2 citrullination induced more dramatic distal changes. Despite distinct PAD2 and PAD4 citrullination patterns, mo-DCs in our

natural antigen processing assay ultimately selected convergent citrullination-dependent repertoires, which suggests that either PAD enzyme could drive the initiation of autoimmunity to citrullinated antigens through the revelation of cryptic epitopes. This finding is in line with our prior observation that ACPAs from RA patients in aggregate do not preferentially target antigens citrullinated by one PAD over the other, but that individual patients can exhibit preferences for PAD2- or PAD4-citrullinated autoantigens, perhaps related to their particular disease etiology[54]. Despite these nuances, our study demonstrates that citrullination results in the generation of a unique peptide repertoire irrespective of the autoantigen or citrullinating PAD enzyme, indicating that a variety of autoantigens or PADs have the potential to trigger autoimmunity to citrullinated antigens in RA.

MHC class II antigen processing employs a complex mechanism, which renders the pathway highly sensitive to the structure of incoming proteins[1,2]. Previous studies have demonstrated that cleavage by a single protease, post-translational modification, or mutation, even at a single site, can lead to a dramatic alteration in protein structure and downstream antigen processing and presentation[3–8]. In fact, these changes can enhance or even be required for the presentation of autoantigenic T-cell epitopes[3–5]. Our study uniquely examines the comprehensive effects of PTM-modulated antigen processing and suggests a link between changes in structure and significant changes in antigen processing induced by citrullination. This

link is supported by the fact that fibrinogen γ, despite lacking citrullination sites of its own, exhibited alterations in antigen processing due to predicted changes in the quaternary structure of fibrinogen following citrullination of the other two chains. In line with findings from previous studies, our data thus support a model in which self-proteins with elevated citrullination potential become modified by PAD enzymes and undergo significant changes in structure and susceptibility to proteolytic cleavage. These changes in antigen processing result in the presentation of a cryptic peptide repertoire and the potential loss of immune tolerance through the activation of autoreactive CD4+ T cells and subsequent initiation of the adaptive immune response. The requisite co-stimulation of CD4+ T cells likely follows the engagement of the innate immune response by environmental stimuli, which leads to the recruitment of immune cells, inflammation-induced cell death, activation of PAD enzymes and elevated production of citrullinated antigens, and the release of inflammatory cytokines. However, based on our current data, we cannot determine whether these autoreactive T cells are directly pathogenic in RA, and future in vivo studies will be needed to determine their role in autoimmunity.

Although our study reveals important principles underlying citrullination-induced modulation of antigen processing, several questions remain to be investigated in future studies. For our cellular assay, we chose to focus on mo-DCs as they represent a powerful and tractable human cellular model system capable of identifying immunologically relevant epitopes[39,55], in addition to the unique relevance of DCs in disease initiation as the primary cell type responsible for activating naive CD4+ T cells. However, mechanisms of antigen uptake into MHC class II processing compartments, along with the regulation of some MHC class II–associated machinery, differ between APC subsets and, in some cases, in response to pro-inflammatory cytokines[56]. A future study comparing the processing of citrullinated autoantigens by a diverse repertoire of APCs and following stimulation by pro-inflammatory cytokines would unveil a comprehensive picture of how citrullination leads to the presentation of disease-relevant epitopes and may identify APC-specific differences in citrullinated autoantigen processing. Further, as MHC class II presentation is quite complex, wherein cathepsin degradation and MHC protection can occur simultaneously, it is important to acknowledge that our study of the biochemical effect of citrullination on antigen processing may exclude potential epitopes by not addressing the contribution of MHC protection. We focused our efforts on the impact of citrullination on processing alone based on previous studies that have shown autoantigen-derived epitopes to be those that survive cathepsin cleavage and rely less heavily on MHC protection and HLA-DM-mediated peptide exchange than pathogen-derived epitopes[21]. However, future investigations should be undertaken to consider the effects of MHC protection and investigate proteases with preferential proteolytic activity toward citrulline- versus arginine-containing cleavage sites to further define how citrullination alters processing at the molecular level. It is known that cathepsin B prefers arginine in the P1 position and that the presence of citrulline at this site reduces proteolytic efficiency[57]. However, we observed increased cleavage directly adjacent to citrulline residues in our proteolytic mapping assay and the destruction of peptides containing a citrullination site in NAPA, suggesting augmented proteolysis by a protease exhibiting preferential cleavage of citrulline, rather than arginine, residues. Finally, the structures utilized in this study were predicted using AlphaFold and may not reflect biological truth, although the observed changes in structure upon citrullination are supported by previous studies[28,58]. As such, further biophysical experiments will be important to validate these structures and the predicted changes in protein structure in response to citrullination.

Our findings enhance the understanding of the loss of immune tolerance in RA, lend support to the crypticity theory of autoimmunity, and broaden the search for disease- and treatment-relevant epitopes. As more peptide-based vaccines and immunotherapies are developed to treat autoimmune diseases, our study emphasizes the risks of restricting the search for tolerizing epitopes to post-translationally modified peptides, rather than taking a more unbiased approach to identify relevant cryptic epitopes. Our data additionally shed light on a fundamental tenet of autoimmunity theory—how small changes to self can have disproportionate impacts on immunologic mechanisms and potentially lead to the dysregulation of the immune response.

## Methods

### Proteolytic mapping by cathepsins B, S, and H

Prior to proteolytic mapping, citrullinated antigens were generated by incubating recombinant human antigens—hnRNP A2/B1 (RA33; custom-made as previously described by in ref. [24]), fibrinogen (Cayman Chemical), and vimentin (PeproTech)—with Sf9-purified peptidylarginine deiminase 2 or 4 (PAD2 or PAD4; Inova) at a 17:1 mass ratio in citrullination buffer (100 mM Tris, 1 mM DTT, 5 mM CaCl$_2$, pH 7.5) for 4 h at 37 °C. Native antigens were diluted in the same buffer without PAD enzymes and without incubation at 37 °C to prevent degradation. Citrullination was confirmed by anti-peptidylcitrulline (clone F95, EMD Millipore) immunoblotting. Native and citrullinated antigen pools were split into four 15 μg replicates per condition before digestion (except hnRNP A2/B1, which was digested in double-volume replicates of three and later split into two each, total $n = 6$, prior to MS) by cathepsins B, S, and H (Millipore Sigma). Digestions were incubated in digestion buffer (citrate phosphate buffer, 4 mM DTT, 1 mM EDTA, pH 5.0) for 4 h at 37 °C. Fibrinogen and vimentin were digested at a 60:1 antigen:cathepsin mass ratio per cathepsin, and hnRNP A2/B1 was digested at a 150:1 antigen:cathepsin mass ratio per cathepsin to achieve full digestion of the protein. Digestion was confirmed by Coomassie Blue staining, and the remaining samples were centrifuged through a 10-kDa-spin filter (Microcon, Millipore) to remove undigested protein fragments. The resulting peptide repertoires were then analyzed by LFQ–MS.

Prior to MS, the samples were reduced and alkylated by incubating in 10 mM Tris (2-Carboxyethyl) phosphine hydrochloride (TCEP, v/v) and 50 mM chloroacetamide (CAA, v/v) for 1 h at room temperature. The samples were then desalted using C$_{18}$ StageTips (3 M Empore) after acidifying with 1% trifluoroacetic acid (TFA). The peptide solution was vacuum-dried and stored at –80 °C. The peptide samples were analyzed on an Orbitrap Fusion Lumos Tribrid mass spectrometer interfaced with an Ultimate 3000 RS Autosampler nanoflow liquid chromatography system (Thermo Scientific). The dried peptides were reconstituted in 15 μL of 0.5% formic acid (FA) and then loaded onto a trap column (Acclaim™ PepMap™ 100 LC C$_{18}$, 5 μm, 100 μm × 2 cm, Thermo Scientific) at a flow rate of 8 μL/min. Peptides were separated on an analytical column (Easy-Spray™ PepMap™ RSLC C$_{18}$, 2 μm, 75 μm × 50 cm, Thermo Scientific) at a flow rate of 0.3 μL/min using a linear gradient with mobile phases consisting of 0.1% FA in water and in 95% acetonitrile (ACN) for 120 min. The mass spectrometer was operated in data-dependent acquisition (DDA) mode. The MS1 (precursor ions) scan range for a full survey scan was acquired from 300 to 1800 $m/z$ in the top speed setting with a resolution of 120,000 at an $m/z$ of 200. The AGC target for MS1 was set as $1 \times 10^6$, and the maximum injection time was 100 milliseconds. The most intense ions with charge states of 2–5 were isolated in a 3-s cycle, fragmented using higher-energy collisional dissociation (HCD) fragmentation with 32% normalized collision energy, and detected at a mass resolution of 50,000 at an $m/z$ of 200. The AGC target for MS/MS (fragments) was set as $5 \times 10^4$, and the ion filling time was 100 milliseconds. The precursor isolation window was set to 1.6 $m/z$ with a 0.4 $m/z$ offset. The dynamic exclusion was set to 30 s, and singly charged ions were rejected. Internal calibration was carried out using the lock mass option ($m/z$ 445.1200025) from ambient air[59–61].

Proteome Discoverer (version 2.4.1.15, Thermo Scientific) was used for the identification and quantitation of proteins. During MS/MS preprocessing, the top 10 peaks in each window of 100 *m/z* were selected for database search. The tandem mass spectrometry data for hnRNP A2/B1 or fibrinogen and vimentin samples were then searched using SEQUEST or MSFragger algorithms, respectively, against a human UniProt database (released in Dec. 2019 or Jan. 2021, respectively), including common contaminant proteins. The search parameters and methods used were as follows: (a) no enzyme (unspecific) including 0 maximum missed cleavage sites; (b) precursor mass error tolerance of 10 ppm; (c) fragment mass error tolerance of 0.02 Da (hnRNP A2/B1) or 20 ppm (fibrinogen and vimentin); (d) carbamido-methylation (+57.02146 Da) at cysteine as fixed modifications; and e) oxidation at methionine (+15.99492 Da), deamidation (+0.98402 Da) at arginine, and acetylation (+42.01057 Da) at protein N-terminus for fibrinogen and vimentin, in addition to methionine loss (−131.04049 Da) at methionine and methionine loss with acetylation (−89.02992 Da) at methionine for hnRNP A2/B1, as variable modifications. The minimum peptide length was set to seven amino acids. Peptides and proteins were filtered at a 1% FDR at the peptide-spectrum match level using the percolator node and at the protein level using the protein FDR validator node, respectively. Protein grouping was performed with a strict parsimony principle to generate the final protein groups. All proteins sharing the same set or subset of identified peptides were grouped, while protein groups with no unique peptides were filtered out. The Proteome Discoverer iterated through all spectra and selected PSM with the highest number of unambiguous and unique peptides. The protein quantification was performed using the area under the curve of precursors with the following parameters and methods. Precursor peak detections were conducted using the Minora Feature Detector node setting the minimum trace length to 5 and max ΔRT (retention time) of isotope pattern multiplets to 0.2 min. Chromatographic alignment and feature mapping were conducted using the Feature Mapper node as follows; RT alignment was enabled, maximum RT shift was set to 10 min, mass tolerance was set to 10 ppm, parameter tuning was set to coarse, RT tolerance for the feature mapping was set to 2 min, mass tolerance for the feature mapping was set to 5 ppm, and minimum signal-to-noise threshold was set to 5. The area under the curve was calculated using the Precursor Ion Quantifier node as follows; both unique and razor peptides were used for the quantification, protein groups were used to consider protein uniqueness, and summed abundances were used for protein abundance calculation. Data normalization was disabled[59–61].

## Citrullination site mapping

Native and citrullinated fibrinogen and vimentin samples were collected post-citrullination reaction during proteolytic mapping, and citrullinated sites were mapped by LFQ–MS. Proteins were reconstituted in 50 μL of urea buffer consisting of 8 M urea in 50 mM triethylammonium bicarbonate (TEAB) and lysed by sonicating (1 min on, 1 min off) for three cycles. The proteins were reduced and alkylated by incubating in 10 mM TCEP (v/v) and 40 mM CAA (v/v) for 1 h at room temperature. The proteins were digested with LysC (Lysyl endopeptidase mass spectrometry grade, Fujifilm Wako Pure Chemical Industries Co.) at 10 ng/μL (v/v) for 3 h at 37 °C. The proteins were further digested with trypsin (sequencing grade modified trypsin, Promega) at 10 ng/μL (v/v) at 37 °C overnight after diluting the urea from 8 M to 2 M with 50 mM TEAB. The digested peptides were desalted using C$_{18}$ StageTips after acidifying with 1% TFA. The peptide solution was vacuum-dried and stored at –80 °C. Peptide samples were analyzed by mass spectrometry as described in the proteolytic mapping Methods section using the SEQUEST algorithm, search parameter (a) set to trypsin as a proteolytic enzyme including two maximum missed cleavage sites, and a minimum peptide length set to six amino acids.

Data from proteolytic mapping and citrullination site mapping MS results were combined when available to detect as many citrullination sites as possible. Citrullination sites were identified from the proteolytic mapping data alone for hnRNP A2/B1. Citrullination sites were filtered on both residue abundance in MS data and the frequency with which a given residue was found to be citrullinated to account for trace citrullination observed in the native samples and identify meaningfully citrullinated residues. Thresholds were optimized by visualizing the number of kept or excluded citrullines at various inclusion criteria (Supplementary Fig. 2d). A combination of thresholds was sought which excluded at least 90% of residues in the native conditions while preserving most citrullinated residues in PAD2 or PAD4 conditions. Any citrullinated residue meeting threshold criteria under trypsinized or antigen processing conditions were incorporated as a citrulline in downstream analysis. In total, 10% citrullination and $1 \times 10^6$ minimum abundance excluded 91.3% of native residues while preserving 72% of citrullination sites observed in the citrullinated samples.

## Identification of changed regions and peptides

Enrichment of each residue in the linear amino acid sequence was calculated for each antigen as the sum of the abundance of all ProtMap-derived peptides containing a given residue in the respective citrullinated sample divided by the sum in the native sample. PAD-enriched, or created, regions were defined as contiguous amino acid residues with log$_2$(enrichment) ≥1, while PAD-reduced, or destroyed, regions were defined as contiguous residues with log$_2$(enrichment) ≤−1.

To define significantly enriched peptides, the abundance of each peptide was compared between 4 and 6 replicates of native versus citrullinated samples by Student's *t* test. Peptides with significantly different abundance (FDR-corrected *P* value (*q* value) ≤ 0.05) and with an ≥twofold enrichment or depletion were considered significantly different between the native and citrullinated groups.

## Proteolytic cleavage site analysis

The presence of exopeptidase activity and peptide trimming in Prot-Map hindered the use of peptide sequence alignment to assess conserved cleavage sites. Instead, putative cleavage sites were identified from log$_2$(enrichment) plots (Fig. 1a). The discrete derivative of log$_2$(enrichment) was used to identify locations of changing differential expression. Local maxima and minima of the derivative exceeding log$_2$(1.3) were identified through the *pracma* R package. This threshold was optimal for the reduction of noise from minor cut sites that minimally impacted abundance. Local extrema were further characterized as novel (created) or masked (destroyed) cut sites based on the relative abundance of residues on either side of the site.

After identifying cleavage sites, the number of citrullines within all P2 to P2' 4-mers was counted. The number of citrullines expected to reside within these residues was obtained via bootstrapping, in which citrulline location was shuffled 1000 times to be at any residue along the sequence length with adequate MS coverage (≥$1 \times 10^6$ abundance). Portions of antigen sequences which were not detected in mass spectrometry data were not considered. In each iteration, the number of citrullines found within P2-P2' cut-site residues was obtained. The expected number of cut-site citrullines was taken as the mean number of cut-site citrullines across all iterations. *P* value was calculated as the fraction of iterations with the same or greater number of citrullines at cut sites as the observed experimental data. Similarly, bootstrapping was applied to determine the expected number of citrullines at any individual cut-site residue between P4 and P4', and *P* value obtained.

## Linear distance to citrulline calculation

Linear distance to citrulline was calculated by taking the minimum distance from the N- or C-terminus of each region to the nearest citrullinated residue in the linear protein sequence. Regions containing

citrulline were assigned distance = 0. The variable length and location of regions inherently alters the probability of being near a citrullinated residue. To compute expected distance to citrulline, bootstrapping was performed in which the positions of known citrullines were shuffled 2000 times to be at any residue along the sequence length. At each iteration, the distance to citrulline was computed. The expected distance to citrulline was calculated as the mean distance across all iterations.

### Antigen processing change score calculation

The antigen processing change score was calculated as the percentage of the total number of peptides identified by ProtMap that were significantly changed between native and citrullinated samples (both enriched/created and reduced/destroyed).

### AlphaFold and protein structure analyses

Predicted native and citrullinated protein structures for vimentin, fibrinogen (β and γ), and hnRNP A2/B1 were generated with Alpha-Fold v2.0 (installed as of commit "1d43aaf" from https://github.com/deepmind/alphafold). FASTAs for native and simulated citrullinated proteins were folded with the AlphaFold Docker script with the following parameters: "–max_template_date = 2020-05-14" and "–db_preset = reduced_dbs" in accordance with recommendations from the AlphaFold documentation. Citrullination at mass spectrometry–confirmed citrullination sites for each protein was simulated by substitution with glutamine residues, which have been historically used to model the non-classical amino acid citrulline, as they possess the same terminal side chain and charge[37,38]. Folded models for each input FASTA with the first rank following Amber relaxation were used for downstream analysis. Predicted structures for the fibrinogen α chain could not be obtained due to computational constraints. Predicted native and citrullinated structures were aligned using the align command in the PyMOL Molecular Graphics System, version 2.4.2 (Schrödinger), and RMSD was calculated using the PyMOL rms command. TM-scores were calculated using the Zhang Lab TM-align online algorithm (https://zhanggroup.org/TM-align/)[32,33]. The locations of autoantigen-derived changed regions from ProtMap were visualized within the AlphaFold-predicted 3D protein structures using PyMOL.

After aligning the modeled native and citrullinated structures for each respective antigen, the positions of all atoms in each structure were exported from PyMOL as.pdb files and analyzed using R[62]. The ($x$, $y$, $z$) coordinates of the α-carbon ($C_a$) of each residue were used to approximate the residue's position. 3D distance to citrulline for each region was calculated as the minimum Euclidean distance between any residue in a given region and any citrulline in the protein. To calculate the expected distance to citrullines via bootstrapping, a number of residues equal to the number of citrullines in each structure were randomly chosen to relabel as residues of interest, and the minimum Euclidean distance for each region to these residues was found. This randomization was repeated 2000 times, and the average minimum distance to the random residues of interest for each region was considered the expected distance to the nearest citrulline.

### Peptide-binding core analysis

The NetMHCII-2.3 peptide-binding affinity prediction algorithm (DTU Health Tech; https://services.healthtech.dtu.dk/service.php?NetMHCII2.3) was used to predict peptide-binding cores to RA-associated SE HLA-DR molecules (HLA-DRB1*01:01, *04:01, *04:04, *04:05, and *10:01)[35]. Mass spectrometry-confirmed citrullination sites were approximated by substituting arginines in the original sequence for glutamine residues. Predicted high-affinity peptide-binding cores were classified as those with predicted affinity <500 nM to any SE allele. The abundance of each high-affinity core was quantified in the aggregate peptide data by summing the abundance of each peptide containing the core sequence, and $\log_2$(enrichment) was calculated for each core. Binding cores with an enrichment of at least 50% were classified as enriched, or created when only present in the respective citrullinated sample, while cores with a reduction of at least 50% were classified as reduced, or destroyed when only present in the native sample. Created cores containing citrulline residues were classified by the position of the citrulline residue as either an MHC class II anchor residue (predicted to bind to pockets P1, P4, P6, or P9) or else as a potential TCR-contact residue. Median binding affinities of total created and destroyed peptide repertoires to SE HLA-DR molecules (*01:01, *04:01, *04:04, *04:05, and *10:01) predicted by NetMHCII-2.3 were compared using nonparametric, two-tailed Mann–Whitney $U$ tests.

### Natural antigen processing assay (NAPA)

Cryopreserved PBMCs from an HLA-DRB1*04:01/16:01-positive (SE heterozygous) healthy leukopak donor were thawed. CD14+ monocytes were positively selected from PBMCs with CD14 MicroBeads (Miltenyi Biotec) and differentiated into mo-DCs for 7 days using Mo-DC Differentiation Medium, containing granulocyte-macrophage colony-stimulating factor (GM-CSF) and interleukin-4 (IL-4; Miltenyi Biotec). In total, $3–10 \times 10^6$ mo-DCs per well in a 6-well plate were incubated overnight at 37 °C in 5% $CO_2$ with 400 μg of either native, PAD2-citrullinated, or PAD4-citrullinated fibrinogen (Cayman Chemical) to allow the mo-DCs to undergo antigen processing and presentation onto HLA-DR molecules. Mo-DCs were then harvested and lysed in cold lysis buffer (1% CHAPS, 1 mM EDTA, and a protease inhibitor cocktail) for 1 h at 4 °C on a rocking table. Insoluble material was removed by centrifugation, and the supernatant was incubated with 20 μg of anti-HLA-DR antibody (1:10 dilution; clone L243, BioLegend) overnight at 4 °C on a rocking table. The antibody:antigen complexes were then incubated with 100 μl Protein G agarose beads (Pierce) for 2 h at 4 °C with gentle mixing. The beads were washed four times with wash buffer (20 mM Tris, 150 mM NaCl, pH 7.4), and the HLA-DR-bound peptides were eluted in 1% TFA with shaking for 15 min at room temperature. This step was repeated once, and the pooled eluate was spun through a C18 Spin Column (Pierce) per the manufacturer's instructions and lyophilized by vacuum centrifugation.

The peptides were then analyzed by MS at the Johns Hopkins Mass Spectrometry and Proteomics Facility[39]. Interfering reagents were removed by strong cation exchange (SCX) on a stage tip, from which peptides were eluted with 400 mM ammonium bicarbonate in 25% ACN and 0.05% formic acid. Samples were dried by vacuum centrifugation, resuspended in 10% ACN and 0.1% formic acid, and diluted in 0.1% formic acid. Samples were injected onto a 2 cm reverse phase trap (YMC ODS-A 10 μm particles, 120 Å; Kyoto, Japan) and separated using a 75 nm × 20 cm reverse phase column (Reprosil 3-μm particles, 100 Å pore size, Ammerbuch-Entringen) with 2–90% ACN gradient in 0.1% formic acid over 90 min at 300 nL/min on an Easy LC nanoLC (Thermo Scientific) interfaced with Q-Exactive Plus mass spectrometer (Thermo Scientific). Peptides were electrosprayed at 2 kV, analyzed at 70 K resolution (MS), and sequenced by collisional dissociation (HCD) with peptide fragments analyzed at 35 K resolution (MS2). Ion target values for MS and MS2 were set at $3 \times 10^6$ and $1 \times 10^5$ with 100 and 150 milliseconds, respectively, and a normalized collision energy of 28. Data files were searched against the 2017 RefSeq83 human database using PEAKS 7 (Bioinformatic Solutions, Inc.) with the following criteria: 5 ppm mass tolerance for peptide mas; 0.02 Da for peptide fragments; oxidation of methionine and deamidation of asparagine, glutamine, and arginine as variable modifications. The database search was performed with no enzyme designation, and search results were filtered at the 1% false discovery rate (FDR) level using the PEAKS decoy-fusion algorithm.

## Peptide–HLA-DR binding affinity analysis

Peptides presented by HLA-DR molecules in NAPA—and when applicable, their citrulline-containing counterparts—were synthesized, and peptide binding to HLA-DRA*01:01/DRB1*04:01 was measured using the proprietary ProImmune REVEAL® assay (Supplementary Table 5). Peptide binding was measured through the stabilization of the peptide-MHC complex, which can be detected by an increase in fluorescent signal following binding of a monoclonal antibody specific for the peptide-MHC complex. Each peptide was given a relative binding affinity score calculated as the percentage of the fluorescent signal generated by an internal positive control peptide. Thus, the relative peptide-binding affinity score increases with affinity for the MHC molecule.

## Patients

Participants in this study were recruited from a longitudinal cohort of rheumatoid arthritis patients at the Johns Hopkins Arthritis Center. All patients participating in this study met the 2010 ACR-EULAR Classification Criteria for Rheumatoid Arthritis or were diagnosed with RA by a board-certified rheumatologist[63]. The study was approved by the Johns Hopkins Institutional Review Board, and all patients provided written informed consent. Patients did not receive compensation for their participation. Demographic characteristics for the participants in this study were obtained from the Johns Hopkins Arthritis Center Longitudinal Registry and are provided in Supplementary Table 6. Healthy control cells were obtained from de-identified leukopaks collected by the Anne Arundel Medical Center. PBMCs were isolated by density-gradient centrifugation (Ficoll-Paque Plus, GE Healthcare) from whole blood for all participants and immediately cryopreserved in Recovery Cell Culture Freezing Medium (ThermoFisher Scientific).

Participants were classified based on SE and/or ACPA status for T-cell stimulation assays. High-resolution HLA-DRB1 genotyping was performed at the Johns Hopkins University Immunogenetics Laboratory by next-generation sequencing from flash-frozen cell pellets. Briefly, HLA-typing was performed with the TruSight HLA Sequencing Panel (Illumina), wherein polymerase chain reaction was used to generate long-range amplicons that were then enzymatically cleaved and end-labeled. Automated paired-end sequencing was performed on the MiSeq System, and the Assign TruSight HLA Analysis software was used to assign HLA alleles. SE+ individuals were designated as those with at least one of the following SE alleles: HLA-DRB1*01:01, *04:01, *04:04, or *04:05. ACPA+ and ACPA− patients were selected based on a review of the clinical record and, when available, CCP3 values collected by the Johns Hopkins Rheumatic Disease Research Core Center (RDRCC). CCP3 values were measured in serum from patients for whom no CCP2 or CCP3 value was available from the clinical record or the RDRCC database (Quanta Lite CCP3 IgG ELISA, Inova Diagnostics).

## T-cell stimulation assays

Cryopreserved PBMCs from ACPA+SE+ RA patients ($n = 10$), ACPA−SE+ RA patients ($n = 8$), and SE+ healthy controls ($n = 10$) were thawed, and $1-1.5 \times 10^6$ cells were plated per well in a 96-well plate in RPMI culture medium supplemented with 5% Human AB serum (Sigma) and allowed to rest for 8 h at 37 °C in 5% $CO_2$. PBMCs were then pre-incubated with 10 μg/ml anti-human CD40 blocking antibody (1:1,300 dilution, G28.5, GeneTex) for 15 min prior to stimulation with 2.5 μM of each candidate NAPA-derived fibrinogen peptide (>95% purity, synthesized by Elim Pharmaceuticals; Supplementary Table 7) or with media alone for 18 h at 37 °C. Following stimulation, cells were stained with 1:100 BV510-conjugated anti-CD3 (UCHT1, BioLegend), 1:80 PacBlue-conjugated anti-CD4 (RPA-T4, BD Biosciences), 1:40 APC-H7–conjugated anti-CD8 (SK1, BD Biosciences), 1:20 PE-conjugated anti-CD154 (MR1, BD Biosciences), 1:80 AF488-conjugated anti-CXCR3 (G025H7, BioLegend),

1:100 PerCP-Cy5.5-conjugated anti-CCR6 (11A9, BD Biosciences), 1:100 PE-Cy7-conjugated anti-CCR4 (1G1, BD Biosciences), and 1:300 Live/Dead Fixable Stain (Molecular Probes).

The percentage of live CD4+ T cells that upregulated CD154 was quantified by flow cytometry (FACSAria II using FACSDiva version 6.0, BD Biosciences) at the Johns Hopkins Bayview Immunomics Core Facility and analyzed using FCS Express 7 version 7.12.0009 (De Novo Software). The upper 95th confidence interval of the mean of responses observed among the healthy controls (CD154+CD4+ T cells ≥0.05865%) was used as a threshold to define positive CD4+ T-cell responses to peptide stimulation. Comparisons between the percentages of all potential T-cell responses for each peptide stimulation group were performed using the $\chi^2$ test. Potential T-cell responses for each peptide and the patient group were defined as the number of patients multiplied by the number of peptides in each group (i.e., created, no change, and destroyed). CD4+ T-helper cell subsets were defined as follows: Th1 (CCR6−CCR4+CXCR3−), Th2 (CCR6−CCR4−CXCR3+), Th17 (CCR6+CCR4−CXCR3+), and Th1−17 (CCR6+CCR4+CXCR3−), using the gating strategy outlined in Supplementary Fig. 7. Paired two-sided Student's $t$ tests were performed to compare the proportions of helper subset phenotypes between CD154+CD4+ T cells and the total T-cell population in each ACPA+ RA patient.

## Tetramer binding assays

MHC class II tetramers containing three of the created NAPA-derived peptides or control peptides were synthesized by ProImmune (Supplementary Table 8). Cryopreserved PBMCs from ACPA+SE+ RA patients ($n = 18$), ACPA−SE+ RA patients ($n = 10$), and SE+ healthy controls ($n = 10$) were thawed and split between tetramer and cytokine secretion assays (see T-cell cytokine secretion analysis method section). In total, $1-1.5 \times 10^6$ cells per condition were then stained for flow cytometry with the MHC class II tetramers (PE- or APC-conjugated) in RPMI culture medium supplemented with 2% fetal bovine serum, 50 nM dasatinib, and 0.1% sodium azide for 2 h at 37 °C in 5% $CO_2$. Following tetramer staining, cells were stained for 30 min at 4 °C in PBS with 50 nM dasatinib and 0.1% sodium azide using the following antibody panel: 1:100 BV510-conjugated anti-CD3 (UCHT1, BioLegend), 1:80 PacBlue-conjugated anti-CD4 (RPA-T4, BD Biosciences), 1:40 APC-H7–conjugated anti-CD8 (SK1, BD Biosciences), 1:80 AF488-conjugated anti-CXCR3 (G025H7, BioLegend), 1:100 PerCP-5.5-conjugated anti-CCR6 (11A9, BD Biosciences), 1:100 PE-Cy7-conjugated anti-CCR4 (1G1, BD Biosciences), 1:80 PE/Cy5-conjugated anti-CD25 (BC96, BioLegend), 1:80 PE/Dazzle 594-conjugated anti-CD127 (A019D5, BioLegend), 1:20 BV650-conjugated anti-CCR7 (G043H7, BioLegend), 1:20 BV605-conjugated anti-CD45RA (M5E2, BioLegend), 1:20 BV711-conjugated anti-PD-1 (EH12.2H7, BioLegend), and 1:300 Live/Dead Fixable Stain (Molecular Probes). Tetramer-positive CD4+ T cells were characterized by flow cytometry (Cytek Aurora using SpectroFlo version 3.0.3, Cytek Biosciences) at the Johns Hopkins Bayview Immunomics Core Facility and analyzed using FCS Express 7 (De Novo Software).

The following CD4+ phenotypes were used: T-regulatory cells (Tregs; CD127loCD25hi), naive T cells (Tnaive; CD45RA+CCR7+), effector memory T cells (Tem; CD45RA−CCR7−), central memory T cells (Tcm; CD45RA−CCR7+), and effector memory RA T cells (Temra; CD45RA+CCR7−), using the gating strategy outlined in Supplementary Fig. 9. Control tetramers were used to exclude cells with non-specific binding from the FibTet+ gate. The upper 95th confidence interval of the mean of responses observed among the healthy controls (CD4+FibTet+ T cells ≥0.5547%) was used as a threshold to define fibrinogen tetramer-positive CD4+ T cells. Comparisons between the frequencies of tetramer-positive cells in each patient group were performed using the $\chi^2$ test, and two-sided Student's $t$ tests were performed to compare the proportions of fibrinogen tetramer-positive CD4+ T-cell effector phenotypes by the patient group.

## T-cell cytokine secretion analysis

In total, $2 \times 10^5$ PBMCs from cryopreserved PBMCs used for tetramer binding assays were plated per condition in a 96-well U-bottom plate and stimulated with 2.5 µM of each NAPA-derived fibrinogen peptide (Supplementary Table 7) or with media alone in RPMI culture medium supplemented with 5% Human AB serum (Sigma). Cells were incubated for 48 h at 37 °C, and the supernatants were harvested for cytokine secretion analysis. Cytokine levels were quantified per the manufacturer's protocol using a Meso Scale Discovery V-PLEX custom assay (Meso Scale Diagnostics; MSD): Pro-inflammatory Panel 1 (human) measuring IFN-γ, TNF-α, IL-4, IL-13, IL-2, and IL-10, and Th17 Panel 1 (human) measuring IL-17A and IL-22. Supernatants were assayed at a 1:2 dilution, and plates were read on a MESO QuickPlex SQ 120MM at the Johns Hopkins Bayview Immunomics Core Facility and analyzed using the MSD DISCOVERY WORKBENCH Desktop Analysis Software version 4.0. Cytokine levels in each sample were normalized by dividing each calculated value by the unstimulated control value for that patient and cytokine. Unstimulated control values equal to zero were set to the lowest detected nonzero value for each cytokine to allow for normalization.

## Statistical analysis

Paired two-sided Student's $t$ tests were performed to compare the abundance of ProtMap-derived peptides between replicates of native and citrullinated samples, and $P$ values were FDR-adjusted by the Benjamini–Krieger–Yekutieli method. Bootstrapping was utilized to calculate the expected number of citrulline residues within the P2-P2′ positions surrounding cleavage sites and within each position (P4-P4′) of citrulline-containing cleavage sites. This method was also applied to calculate the expected linear and Euclidean distances from changed regions to the nearest citrulline residue. Two-tailed Mann–Whitney $U$ tests were performed to compare the median binding affinities of created and destroyed peptide repertoires to SE alleles. Paired or unpaired two-sided Student's $t$ tests were performed to compare proportions of CD4$^+$ helper T-cell subsets and fibrinogen tetramer-positive CD4$^+$ T-cell effector phenotypes, respectively. Finally, two-sided $\chi^2$ tests were used to compare the frequency of T-cell responses and fibrinogen tetramer-positive T cells between patient and/or peptide groups. $P$ values ≤0.05 were considered significant throughout. All statistical analyses were performed using GraphPad Prism, version 9.3.0, or R, version 3.6.3[62].

## Reporting summary

Further information on research design is available in the Nature Portfolio Reporting Summary linked to this article.

## Data availability

The UniProt database (https://www.uniprot.org/) was used for protein identification from the mass spectrometry data. All remaining data generated or analyzed during this study are provided in Supplementary Data 1 and Source Data files. Source data are provided with this paper.

## Code availability

The code used in this study is available at https://github.com/DarrahLab/Curran-et-al-2022_Cit_AgProc (https://doi.org/10.5281/zenodo.7566507)[64].

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

## Acknowledgements

We thank Hong Wang in the Darrah laboratory for technical assistance and support, and Michelle Jones and Marilyn C. Towns (Johns Hopkins Arthritis Center) for the coordination and collection of RA patient samples. We thank John Schroeder (Johns Hopkins School of Medicine) for generously sharing PBMCs from healthy donor leukopaks and Felipe Andrade (Johns Hopkins School of Medicine) for providing the plasmid used to generate hnRNP A2/B1 protein. We also thank Inova Diagnostics for providing purified recombinant PAD2 and PAD4 proteins. This study was supported by a Rheumatology Research Foundation Innovative Research Award (A.C. and E.D.), the Luke Evnin and Deann Wright Fellowship (A.C.), and NIH grant R01-AR079404 (E.D.). The Bayview Immunomics Core was supported by NIH grant P30-AR070254 (C.O.B. and E.D.). The mass spectrometry performed by C.H.N. and Y.J. was supported by the NIH High-End Instrumentation grant S10OD021844 (Y.J. and C.H.N.). The content of this study is solely the responsibility of the authors and does not necessarily represent the official views of the National Institutes of Health.

## Author contributions

A.M.C. and E.D. designed the experiments and wrote the manuscript. A.M.C., Y.J., M.A.T., and J.D.C. performed the experiments with support from E.D. and C.H.N. A.M.C., A.A.G., Y.J., J.D.C., and R.K. performed data and/or computational analyses. A.M.C., A.A.G., M.A.T., J.D.C., and E.D. contributed to data interpretation. A.M.C., A.A.G., and M.A.T. contributed to data visualization. Resources were provided by C.H.N., J.C., and E.D. C.O.B. provided clinical samples and data. All authors reviewed the manuscript. E.D. supervised and conceived the study.

## Competing interests

The authors declare the following competing interests: E.D. is an inventor on a licensed patent (US patent no. 8,975,033) and licensed provisional patent (US patent no. 62/481,158) related to the use of antibodies to PAD3 and PAD2, respectively, in identifying clinically informative disease subsets in RA, and has received consulting fees from Celgene and Bristol Myers Squibb and research support from Pfizer, Celgene, and Bristol Myers Squibb outside of this work. The remaining authors declare no competing interests.
