## [Peer Review File · Nature Communications]

Citrullination modulates antigen processing and presentation by revealing cryptic epitopes in rheumatoid arthritisREVIEWER COMMENTS

Reviewer #1 (expertise in citrullinated antigens, class-II antigen presentation)

This is an excellent report demonstrating that citrullination can alter antigen processing either directly as a result of citrulline residues changing susceptibility to proteases or indirectly due to altered conformation. Of particular interest is that many of the revealed cryptic epitopes do not contain a citrulline residue. This is interesting as it was previously thought that citrulline was a preferred anchor residue for the shared epitope HLA-DR molecules and that this was its primary role in inducing in RA. It is unclear why non citrullinated cryptic epitopes would preferentially bind to the SE haplotype?

The study is weaker when it tried to identify CD4 responses to these cryptic epitopes. The hypothesis was that altered proteolytic digestion creates new epitopes that have not been subject to thymic tolerance. However, they find many responses in normal donors to unaltered epitopes. This implies that many self-epitopes are not subjected to thymic tolerance and casts doubt on their hypothesis. The difference in responses to altered v unaltered epitopes between RA+SE+, RA-SE+, SE+HC is of interest but is unexplained. Why would the RA+ patients lose the responses to unaltered antigens? This may be related to their simplified assay of up regulation of CD154 after overnight stimulation with peptide. This assay may be too sensitive and stimulate weak affinity cross reactive T cells that are unlikely to cause autoimmunity. It is unclear if this assay was repeated multiple times on the same donor and if the responses were always identical. It would have been good to measure affinity to several non citrullinated cryptic epitopes perhaps by multimer binding or longer term restimulation assays and peptide titration. The role of inflammation or IFN γ in this process is also of interest as this cytokine can also alter antigen processing and is a key component of RA. An explanation of the role these cryptic epitopes play in the etiology of RA, from joint erosion, cell death, activation of PAD enzymes, role of citrullinated antigens, macrophages, TNF α , immune complexes, T cells and osteoclasts would be useful.

Reviewer #2 (expertise in structural biology and protein crystallography)

This manuscript discusses an understudied area of research relating to Rheumatoid arthritis, a debilitating affliction affecting a large number of people. It is a welcome window on a disease mechanism that may lead to new therapies. It is also very important that it should reach a wide audience, hence my recommendation that the manuscript be published in Nature Communications.

I had great difficulty reading the manuscript because of two basic flaws:

1- It seems the paper is written backwards. We enter into a lengthy structure prediction section, leading on to epitope presentation in MHC II, culminating in the proposition that citrullination leads to different processing. It sounds a logical cascade of thought, but I suspect the actual cascade was the other way around. The proposition was first stated, as evidenced by some experimental work on processing, leading to the structure prediction to explain the observations. In my opinion, this would be a better arrangement of the paper.

2- There is an over-reliance on AlphaFold predictions of structure. I have experienced the hype in the popular press last year when it was claimed that protein structure is now determined for all known sequences. All scientists know this is not true. AF offers models that may or may not be accurate, verification pending, but are better than any other method of predicting structure. There is a tendency in the text of the manuscript to treat AF models as true structure determinations leading to serious conclusions which might be wide of the mark. I suggest an element of scepticism should be injected, rather than trying to ride the crest of the AF wave. I might be a biased experimentalist.

Additionally, I have great dislike to the repeated use of 'destroyed' or 'created' in reference to epitopes that have been 'masked' or 'revealed'. The former indicates the annihilation or bringing into existence out of thin air some biological entity that was there and now is expunged or bypassing biological cell machinery to bring about something that was not there. The truth of the

matter is much more mundane. The citrullination leads to different processing by cathepsins/proteases that hide some epitopes while making others visible. The epitopes are there all the time, it is only that MHC II can either see new ones or fail to see others that had been visible before.

This leads to another point of contention, in my opinion. The proposition of different processing is quite seductive, and I am a fan. There was a missed opportunity of directing the structural analysis and prediction towards modelling masked and revealed epitopes in cathepsin binding sites instead of concentrating on the downstream of the cascade, the presentation. I recognise this may have to be dealt with on another day, and I wouldn't want the publication of this research to be held up for this purpose. Maybe a note can be added to highlight this point.

My 'criticism' above sounds rather bleak, but I would like to say explicitly that I commend the authors for their sound working methods and diligent attention to detail. I hope my comments above would help make the manuscript a pleasure to read.

There are a few specific points that may need attention, as detailed below.

line 101: Is 'total residues' referring to arginines or all protein residue types? Consequently, how does that differ from 'individual peptides' (line 102)?

line 116: 'Regions' are not clearly defined. Does it mean patches of folded protein or just stretches of part-digested protein peptides? Another opportunity to define 'regions' comes up in para starting at line 574.

line 122: The distal citrulline residues, by sequence, could still be proximal by folding pattern.

line 139: Is 'the linear protein structure' another way of saying 'sequence'?

line 142: 'widespread structural changes' cannot be ascertained without considering the fold. This statement needs to be reconstructed accordingly. Also, line 184 may need similar adjustment.

para starting at line 155: Alphafold v2.0 is a very good tool for predicting protein folds, at least to use as models for 3D structure solution with experimental methods (X-ray diffraction or EM). The predicted models cannot be used deterministically to calculate definitive quantities such as rmsd, as the fold has not been verified. Derived similarity scores may not be reliable.

line 195: Concentrating on peptides binding with high affinity may be a little distraction. Work with MHC I epitopes has shown that higher binding affinity does not correlate with more TCR activation, sometimes it improves, other times it reduces activation. In one case, it led to a different TCR not cross-reactive with the original epitope. This might apply with MHC II as well.

para starting line 209: It might be helpful to the reader if the stats were presented as raw figures as well as percentages, e.g. '25 out 136' as well as e.g. '18%'

para starting at line 319: Here we have an explicit discussion of processing of peptides that bind to MHC II. I suggest moving this paragraph to the beginning of the Discussion section, since there is ample mention of peptide processing before this point, but the processing itself is not discussed until half way down the discussion. I suspect this paragraph could be moved to the introduction even. MHC II presentation follows from peptide processing!

line 349: This statement sounds like a proxy for suggesting the existence of different kinds of cathepsins, or proteases at least. Maybe that should be spelt out more explicitly.

line 353: Searching on google for the word 'crypticity', it said 'did you mean chiropractist'! While it is not difficult to infer its meaning, I think it bears a quick definition.

line 619: The link to the prediction database includes spurious characters. Please correct <https://services.healthtech.dtu.dk/service.php?NetMHCII-2.3>.33 by removing the trailing

characters ').33' from the highlighted link.

Line 768: Legend to Fig 3 does not say if 'native' means experimentally determined or predicted by AF. If experimental, PDB entry codes should be quoted somewhere. Also, the violin plots depict 'the actual minus the expected distance'; is either quantity measured or are they both calculated?

line 868: Extended Data Fig 4; hnRNP A2/B1 model is rather fanciful.

Reviewer #3 (expertise in rheumatoid arthritis)

This was an interesting paper to read postulating that citrullination of proteins shift antigen processing and leads to presentation of non-citrullinated peptides. It is suggested that this could explain break of tolerance in RA.

The hypothesis is not new and in fact the more commonly accepted postulate that citrullinated peptides are presented are circumstantial and controversial. From a more basic point of view it is obvious that the presentation of peptide repertoires are shifted due to enzymatic activity. The strength of the paper is that a correlation of peptides due to citrullination rather than destruction are preferentially presented. However, there are many indirect assumptions. Peptide affinity is based on an algorithm that is far from perfect. The "shared epitope" theory used to clump a number of MHC alleles is not correct and in fact these alleles present very different peptide repertoires. Obviously direct studies of peptide binding to the specific MHC molecules should be done. In addition the structure of the discussed antigens are not crystallised but postulated. The association with RA is very weak as there could be many reasons for a changed repertoire in ACPA-positive individuals, for example different antigen-presenting cells are likely used. Lastly, it is well known peptide binding to MHC in vitro is very poorly correlated with peptide specific tolerance in vivo. If citrullination induces neopeptides in vivo they may also be subjected to tolerisation, it must be investigated how such a mechanism breaks tolerance to make the postulation

RESPONSE TO REVIEWERS' COMMENTS

Reviewer #1 (expertise in citrullinated antigens, class-II antigen presentation)

This is an excellent report demonstrating that citrullination can alter antigen processing either directly as a result of citrulline residues changing susceptibility to proteases or indirectly due to altered conformation. Of particular interest is that many of the revealed cryptic epitopes do not contain a citrulline residue. This is interesting as it was previously thought that citrulline was a preferred anchor residue for the shared epitope HLA-DR molecules and that this was its primary role in inducing in RA. It is unclear why non citrullinated cryptic epitopes would preferentially bind to the SE haplotype?

The study is weaker when it tried to identify CD4 responses to these cryptic epitopes. The hypothesis was that altered proteolytic digestion creates new epitopes that have not been subject to thymic tolerance. However, they find many responses in normal donors to unaltered epitopes. This implies that many self-epitopes are not subjected to thymic tolerance and casts doubt on their hypothesis. The difference in responses to altered v unaltered epitopes between RA+SE+, RA-SE+, SE+HC is of interest but is unexplained. Why would the RA+ patients lose the responses to unaltered antigens? This may be related to their simplified assay of upregulation of CD154 after overnight stimulation with peptide. This assay may be too sensitive and stimulate weak affinity cross reactive T cells that are unlikely to cause autoimmunity. It is unclear if this assay was repeated multiple times on the same donor and if the responses were always identical. It would have been good to measure affinity to several non-citrullinated cryptic epitopes perhaps by multimer binding or longer term restimulation assays and peptide titration. The role of inflammation or IFN γ in this process is also of interest as this cytokine can also alter antigen processing and is a key component of RA. An explanation of the role these cryptic epitopes play in the etiology of RA, from joint erosion, cell death, activation of PAD enzymes, role of citrullinated antigens, macrophages, TNF α , immune complexes, T cells and osteoclasts would be useful.

1. This is interesting as it was previously thought that citrulline was a preferred anchor residue for the shared epitope HLA-DR molecules and that this was its primary role in inducing in RA. It is unclear why non citrullinated cryptic epitopes would preferentially bind to the SE haplotype.

We agree that our findings challenge the existing paradigm that has generally been adopted by the field and helps to reconcile the observations that *in vitro* studies investigating the binding of citrullinated peptides to HLA-DR SE molecules have yielded mixed results, while a dearth of citrulline-containing peptides have been eluted from SE molecules isolated from the RA joint. We have highlighted this in our discussion (line 344)—

"Our findings may thus serve to reconcile the disparate observations that citrulline-containing synthetic peptides exhibit higher binding affinities for SE alleles than unmodified peptides in certain biophysical studies but not others,¹⁻³ and that citrullinated peptides have not been isolated from HLA-DR molecules in the RA synovium despite their proposed role in disease initiation and propagation.^{4,5} The findings of our study and the current paradigm are not by necessity mutually exclusive, and indeed, our study reconciles the seemingly incongruent observations in the literature by revealing native epitopes as potential drivers of autoimmune initiation, perhaps alongside citrulline-containing peptides."

If we accept that citrulline residues within peptides do not consistently enhance binding to SE variants (as seen in the cited studies), then the primary role of SE alleles in RA remains unclear, and our findings do not contradict the current literature. We observed that our citrullination-dependent peptide repertoires from both ProtMap and NAPA were predicted to bind with higher affinity on average to SE alleles than the native-dependent peptide repertoires, suggesting that citrullination, most likely by chance, may create a repertoire of higher affinity

peptides. To validate this finding, we performed a peptide binding study as requested by multiple reviewers to measure the binding affinity of the NAPA-derived peptides using the ProImmune REVEAL[®] assay, which provides a surrogate measurement for affinity (called the relative binding affinity score) based on MHC stabilization via the strength of peptide binding. This experiment confirmed that the citrullination-dependent repertoire binds with higher relative affinity than the native-dependent repertoire *in vitro* (new Fig. 5b). We additionally measured the relative binding affinity score of the native peptides that were destroyed by citrullination and their citrulline-containing counterparts to further address the question of how citrullination directly impacts binding to the SE molecules. Our findings agree with the literature when taken altogether, wherein each native and citrullinated peptide pair exhibited different binding patterns. We observed three distinct outcomes: enhancement, ablation, or no change in binding upon citrullination, further supporting the notion that the relationship between citrullination-specific immune responses and the SE alleles is not as simple as enhanced epitope binding (Fig. 5c, Extended Data Table 6). However, there may in fact be additional mechanisms that result in the association of SE alleles with RA development, such as the recent finding by Ishigaki *et al.* (2022), as highlighted in our discussion (line 359)—

"Intriguingly, a recent study demonstrated that autoimmune-associated HLA alleles may preferentially select autoreactive T cells in the thymus during negative selection,⁶ which could act in conjunction with enhanced binding of cryptic peptides to promote loss of immune tolerance to citrullinated antigens in individuals with predisposing HLA alleles."

2. However, they find many responses in normal donors to unaltered epitopes. This implies that many self-epitopes are not subjected to thymic tolerance and casts doubt on their hypothesis. The difference in responses to altered v unaltered epitopes between RA+SE+, RA-SE+, SE+HC is of interest but is unexplained. Why would the RA+ patients lose the responses to unaltered antigens? This may be related to their simplified assay of upregulation of CD154 after overnight stimulation with peptide. This assay may be too sensitive and stimulate weak affinity cross reactive T cells that are unlikely to cause autoimmunity.

While the classic teaching of central tolerance is that autoreactive T cells are deleted in the thymus, it has now been widely demonstrated in studies by Mark Davis' group at Stanford and others that healthy individuals harbor similar levels of self-reactive and pathogen-specific T cells in their peripheral blood that must be controlled by regulatory T cells in the periphery.⁷⁻¹¹ This may reflect that perfect deletion of even weakly self-reactive clones would create insupportable vulnerabilities in the immune repertoire that foreign pathogens could exploit. Peripheral tolerance and regulatory T cells would be unnecessary in a system with perfect deletion of all potentially autoreactive T cells during central tolerance, and the frequent development of immune-related adverse events mimicking traditional autoimmune diseases in patients receiving checkpoint inhibitor cancer therapy further demonstrates in a human model that these "healthy" (i.e., non-autoimmune) patients harbor autoreactive T cells that are readily activated when mechanisms of peripheral tolerance are impaired.

Further, although we do observe some autoreactive responses in the healthy controls tested, we found significantly fewer responses to any self-peptides in our system by healthy controls than RA patients, and these responses came primarily from a single leukopak donor. Although leukopak donors need to complete screening to qualify as healthy donors for donation, they have not been screened specifically or systematically for autoimmunity. Since autoimmune responses are detectable years to decades before the onset of clinical symptoms, we cannot rule out that any of our SE-positive healthy donors may present with RA in the future. Lastly, unlike the RA patients in our study, the healthy donors are not on immunosuppressive drugs that could dampen autoreactive T cell responses, which may result in a higher background reactivity in healthy controls.

In our experience, the CD154 assay has proven among the most specific for TCR stimulation and among the most specific for patients compared to controls in other studies. We have found that longer term stimulation assays, such as CFSE dilution, lead to even higher background activation in healthy control groups. Additionally, a high sensitivity assay is required to detect autoreactive T cells, as they are low frequency and are more likely to possess low-affinity TCRs, which may contribute to their escape of central tolerance, and are likely activated upon an increase in expression of their cognate self-antigen.¹³⁻¹⁵ In the case of RA, citrullinated antigen expression is significantly higher in patients than in healthy individuals, which likely leads to increased presentation of citrullination-dependent peptides and may thus allow T cell activation despite low affinity interactions.

However, to further address the specificity of the CD154 assay, we performed several new experiments and analyses. First, we have added a differentiation analysis of the T cell stimulation data, which demonstrates that the majority of CD154⁺ cells have a T helper cell phenotype (primarily Th2 and Th17), suggesting that, this method does indeed detect potentially autoreactive T cells that were differentiated in the periphery in response to antigen stimulation in a pro-inflammatory environment (new Fig. 6c). This result is supported by previous findings that CD154-positivity marks effector CD4⁺ T cells with high sensitivity and low background.¹² Second, we measured cytokine secretion in response to peptide stimulation and found that ACPA⁺ patient T cells responding to stimulation with the created peptides secreted Th2- and Th17-type cytokines, which supports our T helper cell phenotyping analysis (new Fig. 6d). Finally, we performed MHC class II tetramer binding assays to further confirm the presence of CD4⁺ T cells specific for citrullination-dependent epitopes and to characterize their effector phenotypes *ex vivo*. We found that ACPA⁺ RA patients have significantly more frequent tetramer-positive cells compared to ACPA⁻ patients (new Fig. 7a-b), and although we detected similar levels of tetramer-positive cells in ACPA⁺ patients and SE⁺ healthy controls, we found that tetramer-positive cells in SE⁺ healthy controls were significantly more likely to be regulatory T cells (new Fig. 7c). Together, our findings are in line with the cited studies above, whereby, although we can detect tetramer-positive T cells to created epitopes in healthy controls as well as ACPA⁺ RA patients, healthy control cells were not previously activated *in vivo* (given their lack of activation upon peptide stimulation alone) and were significantly more likely to be regulatory T cells.

Lastly, ACPA⁻ RA patients responded less frequently to citrullination-dependent peptides and harbored significantly fewer T cells specific for tetramers containing these peptides, supporting our hypothesis that these epitopes may lead to the generation of anti-citrulline immune responses only in ACPA⁺ patients. Their enhanced responses to the native-derived peptide repertoire may simply reflect autoreactivity to native fibrinogen in these patients with active autoimmune disease. Thus far, the antibody responses found in this patient subset have not been well studied. However, the presence of these native-dependent reactivities in ACPA⁻ but not ACPA⁺ patients does not necessarily reflect the "loss" of response to unaltered antigens in ACPA⁺ patients, but more likely represents a divergent pathway to autoimmunity in these two patient groups.

3. It is unclear if this assay was repeated multiple times on the same donor and if the responses were always identical.

In lieu of repeating the assay multiple times on the same patients, we opted to test reactivity of the peptides in several different patients to evaluate the reproducibility of the findings across different individuals. As noted above, this assay has low background and is well-validated to detect antigen-specific T cells, so these responses are not likely to be background events. In addition, this experiment requires $\geq 40 \times 10^6$ PBMCs (at least 40 ml of blood) per patient, and patient PBMCs are a precious resource that we must carefully allocate for several studies/experiments. In response to comments by this reviewer and others, we have performed additional complementary assays with remaining PBMC aliquots to increase the confidence that the response we observe are not occurring by chance. Our MHC class II tetramer and cytokine secretion assays further confirm the presence of these citrullination-dependent epitope-specific T cells as effector cells.

4. It would have been good to measure affinity to several non-citrullinated cryptic epitopes perhaps by multimer binding or longer term restimulation assays and peptide titration.

In our experience, and in the published experience of others in the literature, longer term restimulation assays such as proliferation assays often result in higher background T cell activation due to their ability to stimulate naïve T cells and have higher non-specific activation. We have found the shorter duration CD154 assay to be the most specific for TCR-dependent activation and discriminate the best between patients with disease and healthy controls (who, as stated above, often possess a detectable level of autoreactive T cells). To address this concern and provide further support to the existence of antigen-specific T cells targeting non-citrullinate cryptic epitopes, we performed tetramer assays to measure the frequency of antigen-specific T cells by HLA-DR multimer binding using the non-citrullinated cryptic epitopes derived from NAPA that were used to stimulate T cells in our original experiment.

5. The role of inflammation or IFN γ in this process is also of interest as this cytokine can also alter antigen processing and is a key component of RA.

The focus of the current study is the effect of citrullination on classical antigen processing, as this phenomenon has not yet been investigated. However, we agree that the effect of inflammatory cytokines on processing of citrullinated antigens, in addition to other variables such as APC source, will be interesting to consider in future studies. We added the following note to the discussion (line 409) to address this point:

"However, mechanisms of antigen uptake into MHC class II processing compartments, along with the regulation of some MHC class II-associated machinery, differ between APC subsets and, in some cases, in response to pro-inflammatory cytokines.⁵⁶ A future study comparing the processing of citrullinated autoantigens by a diverse repertoire of APCs and following stimulation by pro-inflammatory cytokines would unveil a comprehensive picture of how citrullination leads to the presentation of disease-relevant epitopes and may identify APC-specific differences in citrullinated autoantigen processing."

6. An explanation of the role these cryptic epitopes play in the etiology of RA, from joint erosion, cell death, activation of PAD enzymes, role of citrullinated antigens, macrophages, TNF α , immune complexes, T cells and osteoclasts would be useful.

While it is not within the scope of the current study to explain all of the listed components of RA pathogenesis, we have added a note to the discussion (line 395) in order to address the role of these cryptic epitopes within the broader context of RA:

"In line with findings from previous studies, our data thus support a model in which self-proteins with elevated citrullination potential become modified by PAD enzymes and undergo significant changes in structure and susceptibility to proteolytic cleavage. These changes in antigen processing result in the presentation of a cryptic peptide repertoire and the potential loss of immune tolerance through the activation of autoreactive CD4⁺ T cells and subsequent initiation of the adaptive immune response. The requisite co-stimulation of CD4⁺ T cells likely follows the engagement of the innate immune response by environmental stimuli, which leads to the recruitment of immune cells, inflammation-induced cell death, activation of PAD enzymes and elevated production of citrullinated antigens, and the release of inflammatory cytokines".

Reviewer #2 (expertise in structural biology and protein crystallography)

This manuscript discusses an understudied area of research relating to Rheumatoid arthritis, a debilitating affliction affecting a large number of people. It is a welcome window on a disease mechanism that may lead to new therapies. It is also very important that it should reach a wide audience, hence my recommendation that the manuscript be published in Nature Communications.

I had great difficulty reading the manuscript because of two basic flaws:

1. It seems the paper is written backwards. We enter into a lengthy structure prediction section, leading on to epitope presentation in MHC II, culminating in the proposition that citrullination leads to different processing. It sound a logical cascade of thought, but I suspect the actual cascade was the other way around. The proposition was first stated, as evidenced by some experimental work on processing, leading to the structure prediction to explain the observations. In my opinion, this would be a better arrangement of the paper.

We completely agree and apologize if the current flow of the paper was unclear. The layout of our paper is actually already what you described. We begin with our proteolytic mapping/antigen processing assay, whereby we digest autoantigens with cathepsins BSH and analyze the resulting peptide repertoires in the presence or absence of citrullination in Fig. 1 (i.e., our experimental work on processing). At this point, we have not yet done structural analysis, but rather consider where the resulting peptides originate in the linear protein sequence. We conclude early in the first section that "Citrullination alters antigen processing, resulting in the simultaneous generation of cryptic peptides and destruction of previously dominant peptides." Following this observation that citrullination does lead to altered processing, we then directly investigate the effect of citrullination on cathepsin cleavage activity and changes in structure to explain the differences we see in processing in Fig. 2-3. Thus, we observe changes in processing prior to studying changes in structure, as you suggested. This all occurs before we investigate epitope presentation on MHC class II molecules in Fig. 4-5.

We have clarified in the text where in the study we were examining the linear protein sequence using our ProtMap versus the AF predicted protein structures, and we sought to specify more clearly when our purpose was to investigate antigen processing versus presentation.

2. There is an over-reliance on AlphaFold predictions of structure. I have experienced the hype in the popular press last year when it was claimed that protein structure is now determined for all known sequences. All scientists know this is not true. AF offers models that may or may not be accurate, verification pending, but are better than any other method of predicting structure. There is a tendency in the text of the manuscript to treat AF models as true structure determinations leading to serious conclusions which might be wide of the mark. I suggest an element of scepticism should be injected, rather than trying to ride the crest of the AF wave. I might be a biased experimentalist.

We appreciate your insight as a structural biologist and recognize the limitations of AlphaFold for predictions of protein structure. Citrullinated proteins have not been crystallized to date, and most of the proteins used in this study do not have complete crystal structures available, even for their native forms. Further, as citrullination results in protein unfolding and destabilization, it may pose unique challenges to crystallization. Since we only had partial native structures publicly available and knowing that the different methods (crystallization vs. AlphaFold prediction) would likely generate nonidentical structures, we predicted both the native and citrullinated structures for each antigen to maintain consistency and report only the changes introduced by citrullination within the AlphaFold system.

For this paper, the advancements in *in silico* protein structure prediction realized by AlphaFold allow us to address the potential impact of structural changes on alterations in antigen processing where we otherwise could not. While we cannot be sure that the structures predicted by AlphaFold represent the “true” biological structure of native and citrullinated proteins *in vivo*, since it is nearly impossible to know this with absolutely certainty using any currently available *in vitro* or *in silico* method, we have confidence that the algorithm is revealing how citrullination may change protein structure within the *in silico* universe and reveals common principles between antigens. Further, Nature Communications has published articles that utilized AlphaFold to support their observations without additionally performing experimental structural determinations.^{16–18} Based on your advice, we’ve added caveats throughout the text (*e.g.*, on lines 176 and 429) to clearly express the limitations of these models.

3. Additionally, I have great dislike to the repeated use of 'destroyed' or 'created' in reference to epitopes that have been 'masked' or 'revealed'. The former indicates the annihilation or bringing into existence out of thin air some biological entity that was there and now is expunged or bypassing biological cell machinery to bring about something that was not there. The truth of the matter is much more mundane. The citrullination leads to different processing by cathepsins/proteases that hide some epitopes while making others visible. The epitopes are there all the time, it is only that MHC II can either see new ones or fail to see others that had been visible before.

The authors understand your dislike of the words “created” and “destroyed,” but these were chosen for ease of reading comprehension and to provide a clear distinction between the two groups of peptides. The concept of dominant epitopes being “destroyed,” facilitating the presentation of otherwise cryptic epitopes, is language used by Eli Sercarz and colleagues in the 1990s to describe how autoimmunity could result from recognition of previously hidden parts of a protein. Although the epitopes do always exist within the intact protein sequence, we use these terms to refer to the generation or loss of individual peptides, liberated from the intact protein, which may genuinely be created or destroyed as a peptide entity.

We have added definitions in the main text at their first use to clarify their intended meaning in this study as descriptors of the fates of these epitopes (line 101).

4. This leads to another point of contention, in my opinion. The proposition of different processing is quite seductive, and I am a fan. There was a missed opportunity of directing the structural analysis and prediction towards modelling masked and revealed epitopes in cathepsin binding sites instead of concentrating on the downstream of the cascade, the presentation. I recognise this may have to be dealt with on another day, and I wouldn't want the publication of this research to be held up for this purpose. Maybe a note can be added to highlight this point.

We agree that MHC class II presentation is quite complex, wherein cathepsin degradation and MHC protection can occur simultaneously, and it's possible that we're missing some potential epitopes by not addressing the contribution of MHC protection. However, it is still important to note that in Dr. Sadegh-Nasseri's cell-free MHC class II processing and presentation assay, autoantigen-derived epitopes are those that survive cathepsin cleavage and rely less on MHC protection and DM-mediated peptide exchange than pathogen-derived epitopes. Thus, we believe our processing assay provides important insight directly into changes in antigen processing alone upon citrullination, and likely represents a comprehensive potential peptide repertoire from which MHC class II molecules will select peptides. At your suggestion, we have added a note to address this limitation in the discussion (line 415).

My 'criticism' above sounds rather bleak, but I would like to say explicitly that I commend the authors for their sound working methods and diligent attention to detail. I hope my comments above would help make the manuscript a pleasure to read.

There are a few specific points that may need attention, as detailed below.

5. line 101: Is 'total residues' referring to arginines or all protein residue types? Consequently, how does that differ from 'individual peptides' (line 102)?

'Total residues' refers to all amino acid residues in the protein, illustrating that the peptide repertoire changes significantly despite such a small percentage of the protein sequence changing by post-translational modification. We clarified this in the text by referring to the protein sequence rather than total amino acid residues. 'Individual peptides' refers to the peptides generated by proteolytic mapping (digestion with cathepsins B, S, and H). We added further explanations to the text to clarify how changes in abundance at each amino acid were calculated and differ from changes among individual peptides.

6. line 116: 'Regions' are not clearly defined. Does it mean patches of folded protein or just stretches of part-digested protein peptides? Another opportunity to define 'regions' comes up in para starting at line 574.

Changed regions refer to any stretch of contiguous amino acid residues that was enriched with $\log_2(\text{enrichment}) \geq 1$ in either the native or citrullinated samples, which can be visualized as peaks exceeding $y = 1$ or $y = -1$ in Fig. 1a. We expanded the definition given on the specified line (line 123) to improve clarity.

Changed regions were also defined in the methods in the section titled "Identification of changed regions and peptides", prior to the original line 574. From the text (lines 628-632)—

"Enrichment of each residue in the linear amino acid sequence was calculated for each antigen as the sum of the abundance of all ProtMap-derived peptides containing a given residue in the respective citrullinated sample divided by the sum in the native sample. PAD-enriched, or created, regions were defined as contiguous residues with $\log_2(\text{enrichment}) \geq 1$, while PAD-reduced, or destroyed, regions were defined as contiguous residues with $\log_2(\text{enrichment}) \leq -1$."

7. line 122: The distal citrulline residues, by sequence, could still be proximal by folding pattern.

Agreed. We address this possibility in Fig. 3c, wherein we performed the same analysis of distance to nearest citrulline using the Euclidean distance within the predicted protein structure. We find the same pattern of distal citrullines even accounting for the folding pattern. From the text (lines 192-201)—

"To investigate the correlation between changed regions and citrulline residues in the 3-dimensional (3D) protein structure, we calculated the Euclidean distance between each changed region and the nearest citrulline residue, relative to the expected distance. Similar to the pattern observed for linear distances of changed regions to citrullines (Fig. 2c), we found both citrulline-proximal and citrulline-distal changed regions and observed different distributions between created and destroyed regions, with created regions more frequently distal to citrullines and destroyed regions more frequently proximal (Fig. 3c). This finding is consistent with the higher frequency of citrullines within destroyed regions (Fig. 2a). Furthermore, the persistence of changed regions distal to citrulline residues in the 3D analysis emphasizes the role of citrullination in inducing widespread structural changes that can impact antigen processing from afar."

8. line 139: Is 'the linear protein structure' another way of saying 'sequence'?

That is correct. Apologies for the confusing language. We changed it on line 94 of the text to 'linear protein sequence' for clarity.

9. line 142: 'widespread structural changes' cannot be ascertained without considering the fold. This statement needs to be reconstructed accordingly. Also, line 184 may need similar adjustment.

We removed the qualifier 'widespread' in both cases, as we simply wanted to convey that changes in processing occurring far from citrulline residues suggests that citrullination-induced changes in structure are responsible for these far-reaching/distal impacts on processing.

10. para starting at line 155: AlphaFold v2.0 is a very good tool for predicting protein folds, at least to use as models for 3D structure solution with experimental methods (X-ray diffraction or EM). The predicted models cannot be used deterministically to calculate definitive quantities such as rmsd, as the fold has not been verified. Derived similarity scores may not be reliable.

As acknowledged above, we recognize that AlphaFold has important limitations and have aimed to inject ample skepticism throughout the manuscript to that effect. However, our use of RMSD here is to provide quantification for the structural change induced by citrullination within the *in silico* universe of AlphaFold, not necessarily to say with absolute certainty that this exact number reflects structural changes *in vivo*. To that effect, structures and RMSD values obtained using any experimental method which relies on recombinant expression of a protein in a non-physiological expression system (*e.g.*, *E. coli*), often missing key domains due the inability to obtain structure on unstructured domains or linker regions, could be viewed with an equal level of skepticism. In addition, these values have the most utility as reporting on relative differences between structures obtained using the same methods, as we did here, rather than between methods, even experimentally defined methods (*e.g.*, would be less reliable to compare structures obtained with X-ray diffraction and cryo-EM).

11. line 195: Concentrating on peptides binding with high affinity may be a little distraction. Work with MHC I epitopes has shown that higher binding affinity does not correlate with more TCR activation, sometimes it improves, other times it reduces activation. In one case, it led to a different TCR not cross-reactive with the original epitope. This might apply with MHC II as well.

Here we are interested in peptides predicted to bind to RA-associated HLA-DR molecules to get a sense of which ProtMap-generated peptides would ultimately be selected and bind to DR molecules following processing. We are not focusing on peptides that bind with < 500 nM affinity to predict TCR activation directly. Since the current paradigm holds that citrulline-containing peptides bind preferentially to SE molecules, we sought to interrogate whether, from all peptides generated by antigen processing, the most predicted putative binding cores belong to citrullination-enriched peptides comprised of native sequences or citrulline-containing cores. However, we are using a generous cut-off for putative binders to predict any cores that could potentially bind MHC class II. This cut-off thus encompasses "weakly binding" epitopes that can still potentially activate TCRs. For example, most of the peptides that we isolated directly from MHC class II molecules in NAPA have predicted binding affinities to DRB1*04:01 higher than our 500 nM cut-off, all of which were able to stimulate T cells.

12. para starting line 209: It might be helpful to the reader if the stats were presented as raw figures as well as percentages, *e.g.* '25 out 136' as well as *e.g.* '18%'

We agree that this would be helpful to readers and have added the raw figures to the text (line 228) as you suggested.

13. para starting at line 319: Here we have an explicit discussion of processing of peptides that bind to MHC II. I suggest moving this paragraph to the beginning of the Discussion section, since there is ample mention of peptide processing before this point, but the processing itself is not discussed until half way down the discussion. I suspect this paragraph could be moved to the introduction even. MHC II presentation follows from peptide processing!

The topic of antigen processing has been discussed at length by the discussion, as you mentioned, and our discussion is focused more on contextualizing our results within the scope of current literature than re-summarizing the sequential events of antigen processing and presentation. The first few sentences of the introduction do closely mirror the beginning of the paragraph you referenced to ensure that processing and presentation are introduced early and in the correct order.

14. line 349: This statement sounds like a proxy for suggesting the existence of different kinds of cathepsins, or proteases at least. Maybe that should be spelt out more explicitly.

All cathepsins have unique specificities with preferences for cleavage after different amino acids. Here we suggest that at least one of the proteases involved in antigen processing in our systems prefers to cleave after citrulline residues. The identity of this protease cannot be confirmed at this point, as cathepsin cleavage sites have only been mapped for canonical amino acids, but this does not require a new kind of cathepsin.

15. line 353: Searching on google for the word 'crypticity', it said 'did you mean chiropracist'! While it is not difficult to infer its meaning, I think it bears a quick definition.

As suggested, we have explicitly defined 'crypticity' in the text. As mentioned above, the concept of dominant and cryptic epitopes, and epitope crypticity, was coined by Eli Sercarz, Gilles Benichou, and colleagues in the early 1990s to describe how autoimmunity could result from recognition of previously hidden parts of a protein.

16. line 619: The link to the prediction database includes spurious characters. Please correct <https://services.healthtech.dtu.dk/service.php?NetMHCII-2.3>.33 by removing the trailing characters '.33' from the highlighted link.

We appreciate your attention to detail. The number '33' is the in-text citation for NetMHC. It's active and not part of the link in the Word document, but the sentence has been rearranged to avoid its inclusion in the link in the PDF version.

17. Line 768: Legend to Fig 3 does not say if 'native' means experimentally determined or predicted by AF. If experimental, PDB entry codes should be quoted somewhere. Also, the violin plots depict 'the actual minus the expected distance'; is either quantity measured or are they both calculated?

Native refers to structures predicted from native sequences by AlphaFold. In the Fig. 3 legend, we state that we performed structural alignment of native and citrullinated PDB structures predicted by the AlphaFold AI system. These antigens have not all been fully crystallized, and for consistency, we predicted both native and citrullinated sequences. On line 677 in the Methods, we state that both native and citrullinated protein structures were predicted by AlphaFold, which is why we didn't cite PDB entry codes.

The actual distance between citrulline residues and changed regions was calculated as the minimum Euclidean distance between any residue in a given region and any citrulline in the protein from the predicted atomic coordinates of each amino acid residue. The expected distance to citrullines was calculated via bootstrapping, where a number of residues equal to the number of citrullines in each structure were randomly

chosen to relabel as residues of interest and the minimum Euclidean distance for each region to these residues was found. These calculations are both described starting on line 693 in the Methods.

18. line 868: Extended Data Fig 4; hnRNP A2/B1 model is rather fanciful.

We agree, but as you know, these unstructured regions also cannot be assessed by x-ray crystallography and cryo-EM due to their intrinsically disordered nature. This structure is just one possible fold predicted by AlphaFold. We've added the following note to line 176 of the results to make it clear that these unstructured domains are predicted with low confidence:

"Similarly to experimental methods commonly used to determine protein structure, AlphaFold predicts the conformation of unstructured or intrinsically disordered protein domains with low confidence; thus, predicted misalignment between native and citrullinated proteins in these domains are more likely to be driven by stochastic differences in conformational predictions."

Reviewer #3 (expertise in rheumatoid arthritis)

This was an interesting paper to read postulating that citrullination of proteins shift antigen processing and leads to presentation of non-citrullinated peptides. It is suggested that this could explain break of tolerance in RA.

The hypothesis is not new and in fact the more commonly accepted postulate that citrullinated peptides are presented are circumstantial and controversial. From a more basic point of view it is obvious that the presentation of peptide repertoires are shifted due to enzymatic activity.

The strength of the paper is that a correlation of peptides due to citrullination rather than destruction are preferentially presented. However, there are many indirect assumptions. Peptide affinity is based on an algorithm that is far from perfect. The "shared epitope" theory used to clump a number of MHC alleles is not correct and in fact these alleles present very different peptides repertoire. Obviously direct studies of peptide binding to the specific MHC molecules should be done. In addition the structure of the discussed antigens are not crystallised but postulated. The association with RA is very weak as there could be many reason for a changed repertoire in ACPA-positive individuals, for example different antigen-presenting cells are likely used. Lastly, it is well known peptide binding to MHC in vitro is very poorly correlated with peptide specific tolerance in vivo. If citrullination induces neoepitopes in vivo they may also be subjected to tolerisation, it must be investigated how such a mechanism break tolerance to make the postulation.

1. The hypothesis is not new and in fact the more commonly accepted postulate that citrullinated peptides are presented are circumstantial and controversial. From a more basic point of view it is obvious that the presentation of peptide repertoires are shifted due to enzymatic activity.

As we acknowledge in our manuscript, our hypothesis is indeed built on studies demonstrating the impact of small changes on proteolytic cleavage and the potential role of cryptic epitopes in the initiation and pathogenesis of autoimmunity, but this hypothesis has not been systematically evaluated in RA. Rather, as you note and as noted by Reviewer 1, the more commonly accepted postulate in the field of RA and the one largely accepted as dogma, is the hypothesis that citrullinated peptides are the drivers of the autoimmune response through enhanced binding to shared epitope alleles. Due to the widespread acceptance of this paradigm, even in the face of contradictory in vitro studies demonstrating that citrullinated peptides often do not bind better to SE alleles than their native counterparts and a paucity of citrullinated peptides that have been eluted from HLA-DR alleles in the RA joint, our current study is necessary. Our data may be logical and seem obvious to immunologists versed in the idea of crypticity, but there is a fundamental gap in research regarding this mechanism, which has left a void for assumption that must be rectified to prevent further research and therapeutic strategies focusing merely on citrullinated epitopes as the cause of RA.

2. Peptide affinity is based on an algorithm that is far from perfect. The "shared epitope" theory used to clump a number of MHC alleles is not correct and in fact these alleles present very different peptides repertoire. Obviously direct studies of peptide binding to the specific MHC molecules should be done.

Although we agree and understand the limitations of the MHC class II prediction algorithms, our proteolytic mapping assay generates too many peptides to perform direct binding studies on the entirety of the peptide repertoires. We thus took advantage of the binding algorithms to predict likely binding cores more accurately and efficiently than could be done manually, which allowed us to group sets of peptides derived from the same protein regions and investigate overarching changes in the complete repertoire. However, to investigate the question posed by multiple reviewers regarding the study of peptide binding affinity, we experimentally measured the relative binding affinity of the native, citrullination-dependent peptides derived from NAPA, where presentation is the direct result of the cellular antigen processing machinery (rather than prediction), to HLA-

DRB1*04:01 using the ProImmune REVEAL[®] assay, which provides a surrogate measurement for binding affinity (called the relative binding affinity score) based on MHC stabilization via the strength of peptide binding. These data confirmed that the citrullination-dependent repertoire binds with higher relative affinity than the native-dependent repertoire (new Fig. 5b). We also compared the relative affinity scores of these native peptides to their citrulline-containing counterparts that were destroyed in our assay, to further illuminate the mechanisms behind the changing peptide repertoire. We observed three distinct outcomes upon citrullination of these peptides: enhancement, ablation, or no change in binding (new Fig. 5c); these findings corroborate the current literature when taken in aggregate, in which peptide citrullination has been shown to affect the binding to SE alleles differently depending on the peptides tested.

Of note, our MHC class II binding prediction analysis does not rely on the validity of the shared epitope hypothesis or the similarity of the peptide repertoires bound by different alleles. We independently performed the prediction algorithm for the entire peptide repertoire for each citrullination condition to each of the SE alleles available in the algorithm: *01:01, *04:01, *04:04, *04:05, and *10:01. Putative peptide binders were considered those peptides predicted to bind to any of the SE variants with < 500 nM affinity, rather than limiting our analysis to peptides that were predicted to bind with high affinity to multiple SE molecules. Our analysis simply relies on the well-established association between RA and these particular MHC class II alleles, which renders them relevant molecules to study in the context of disease initiation and pathogenesis. However, to confirm the validity of our approach, we visualized the distribution of putative binding cores between categories representing the behavior of each core in response to citrullination independently for each SE allele (new Extended Data Fig. 5c). The phenomenon we report persists whether putative binding cores were aggregated for all SE alleles or analyzed independently for each allele, and a note was added to this effect on line 232.

Importantly, the phenomenon of preferential presentation of native, citrullination-dependent peptides that we observe utilizing the prediction algorithm is confirmed in a direct cellular system by NAPA. There too we find the presentation of a unique, citrullination-dependent peptide repertoire comprised only of native peptides.

3. In addition the structure of the discussed antigens are not crystallised but postulated.

We recognize the limitations of AlphaFold for predictions of protein structure. However, citrullinated proteins have not been crystallized to date, and most of the proteins used in this study do not have complete crystal structures even in their native forms. Further, as citrullination results in protein unfolding, it may pose unique challenges to crystallization.

For our current study, the advancements in *in silico* protein structure prediction realized by AlphaFold allow us to address the potential impact of structural changes on alterations in antigen processing where we otherwise could not. Based on the advice of the reviewers, we've added caveats throughout the text (*e.g.*, on lines 176 and 429) to clearly express the limitations of these models.

4. The association with RA is very weak as there could be many reason for a changed repertoire in ACPA-positive individuals, for example different antigen-presenting cells are likely used.

We agree that the mechanism we identified for how citrullination may reveal cryptic native epitopes could co-occur with other mechanisms. To our knowledge, there have been no studies that have addressed whether the presented peptide repertoire in ACPA-positive RA patients differs from that of ACPA-negative RA patients or healthy controls. In addition, we are not aware of any data showing differential presentation of the RA autoantigen peptide repertoire by different APCs. As such, in the absence of data, one can speculate that any number of variables may be driving an altered repertoire. However, in the presence of our current data, it is reasonable to conclude that altered processing and presentation of citrullinated antigens and revelation of a

novel citrullination-dependent peptide repertoire is likely a contributing factor. As we mention in the discussion, we too believe that different APCs, such as conventional DCs and B cells, may be contributing to the process, and a comparison of the processing of citrullinated antigens by a diverse repertoire of APCs could reveal further disease-relevant epitopes and APC-specific mechanisms of altered antigen processing (e.g., via protection of citrullinated sites by ACPAs during B cell processing). We, therefore, look forward to data forthcoming by our group and likely others on this topic.

5. Lastly, it is well known peptide binding to MHC *in vitro* is very poorly correlated with peptide specific tolerance *in vivo*. If citrullination induces neoepitopes *in vivo* they may also be subjected to tolerisation, it must be investigated how such a mechanism break tolerance to make the postulation.

To address this point, we performed further analysis of RA patient T cells specific for the citrullination-dependent repertoire. We can detect CD4⁺ T cells in ACPA⁺ RA patients that are activated in response to stimulation with this unique peptide repertoire at a higher level than the native-dependent repertoire and than healthy and ACPA⁻ controls (Fig. 6a-b). These cells upregulate CD154 during a single overnight stimulation with peptide alone, suggesting that they were previously activated *in vivo* and susceptible to restimulation without the addition of signal 2. Phenotype analysis of these T cells also reveals that the majority of CD154⁺ cells coexpress surface markers of effector T helper cells (primarily Th2 and Th17-type), further suggesting that these cells have not been tolerized *in vivo* (Fig. 6c).

Additionally, we performed new experiments to address whether these citrullination-induced neoepitopes may have been subject to tolerization *in vivo*. First, we measured cytokine secretion in response to peptide stimulation and found that ACPA⁺ patient T cells responding to stimulation with the created peptides secrete Th2- and Th17-type cytokines, which supports our T helper cell phenotyping analysis and suggests that these cells have not been anergized (*i.e.*, unresponsive to peptide stimulation without the addition of IL-2) *in vivo* (new Fig. 6d). Second, we performed MHC class II tetramer binding assays characterize the effector status and phenotype of CD4⁺ T cells specific for citrullination-dependent epitopes *ex vivo*. We found that ACPA⁺ RA patients have significantly more frequent tetramer-positive cells compared to ACPA⁻ patients (new Fig. 7a-b), and that tetramer-positive cells in ACPA⁺ patients were significantly more like to be conventional effector memory cells than SE⁺ healthy controls, who had similar levels of tetramer-positive precursors but these were significantly more likely to be regulatory T cells (new Fig. 7c). Thus, as these cells have not been deleted via activation-induced cell death/Fas-FasL engagement nor anergized *in vivo*, in addition to displaying T helper and effector phenotypes, we can provide ample evidence that these T cells were not subjected to tolerization *in vivo*.

REVIEWERS' COMMENTS

Reviewer #3 (expert in rheumatoid arthritis):

The revision have improved the paper. I have just one comment regarding the last point of my comment i.e. they have too strong claims that tolerisation is induced in T cells to the neo-epitope. They added considerable new data. However, the significance of the data is somewhat unclear to me. Fig 6a looks very interesting and instructive but they should calculate the relevant significance and indicated. If it is significant (which it looks like) they should keep this figure in the paper. The other figures 6b-c and fig 7 have important information but really difficult to make definite conclusions from, maybe it could keep this as supplement. The data should be described better, clearly pointing out that it is assumptions when they refer to data that show no proven (significant) difference, but could nevertheless be important information. And then refer to the new data and then address the question of whether T cells are tolerized to neo-epitope somewhat more modestly.

These are just suggestions to improve the paper.

Reviewer #3 acting as mediator for comments from Reviewer 1 (absent; expertise in citrullinated antigens, class-II antigen presentation):

It is correct that autoreactive T cells reactive with unmodified and commonly occurring peptide peptides are frequent in the repertoire. But I guess this is the point, that's the reason these are easily picked up in this assay. The question is which function they have. There is no obvious reason to think they are pathogenic, rather they may have no role (ignorant) or to be regulatory. There is no way to show this in the present work, but the writing should take such considerations into account. The only way to really show functional effects is in corresponding experimental models and data in models of RA shows that there are very specific T cells that actually could mediate arthritis, and these are uniquely identifying posttranslational (glycosylation) modified peptides (on collagen II which is also a citrullination target) and are in fact not completely tolerized in thymus. Such T cell reactivities are conserved between mice and humans and the authors hypothesis is compatible with a role for such T cells. Thus, pathogenic T cells are likely to be exceptions.

Having this said, I think with the restrictions the paper has, it will not be possible to make a better experimental work than they have already done, although I think they could improve the discussion maybe including some of the points above.

RESPONSE TO REVIEWERS' COMMENTS

Reviewer #3 (expert in rheumatoid arthritis)

The revision have improved the paper. I have just one comment regarding the last point of my comment i.e. they have too strong claims that tolerisation is induced in T cells to the neo-epitope. They added considerable new data. However, the significance of the data is somewhat unclear to me. Fig 6a looks very interesting and instructive but they should calculate the relevant significance and indicated. If it is significant (which it looks like) they should keep this figure in the paper. The other figures 6b-c and fig 7 have important information but really difficult to make definite conclusions from, maybe it could keep this as supplement. The data should be described better, clearly pointing out that it is assumptions when they refer to data that show no proven (significant) difference, but could nevertheless be important information. And then refer to the new data and then address the question of whether T cells are tolerized to neo-epitope somewhat more modestly. These are just suggestions to improve the paper.

We appreciate your thoughtful comments and assessment of our manuscript. In order to alleviate the concern that some of our conclusions come across too strong, we have made several changes to the manuscript explained as follows.

Fig. 6a is a graphical representation of the flow cytometry data demonstrating the percentage of CD4⁺ cells that upregulate CD154 in response to peptide stimulation to show the underlying data. Because there are different numbers of peptides between groups, we cannot directly perform statistics to compare them in this format. Our statistical analysis of this data comes in Fig. 6b when we compare the proportion of each patient group that possess peptide response T cells, which is based on the positivity threshold shown in Fig. 6a. From Fig. 6b and 6c, we can conclude that significantly more ACPA+ RA patients harbor T cells specific for the cryptic peptides we identified than ACPA- RA patients and healthy donors do, in addition to the fact that these T cells tend to show enrichment of Th2 and Th17 phenotypes—this is in line with reported CD4 T cell subsets implicated in RA.

We have added statistics as requested to Fig. 7a, which we agree has improved the clarity and presentation of our data. Paired with Fig. 7b, we can conclude that fibrinogen tetramer positive cells are more frequent in ACPA+ RA patients and healthy controls than in ACPA- RA patients with a divergent disease course. Interestingly, we then find that these tetramer positive cells are significantly more likely to be Tregs in healthy controls, which fits with a wider body of literature that demonstrates healthy controls harbor similar levels of T cells with autoreactive TCRs but that they are subject to tolerance mechanisms. This is further supported by the fact that we find tetramer positive cells in healthy donors, but we don't find similar levels of activated cells when we restimulate *ex vivo* with the same peptides, as this assay is designed to restimulate memory T cells, but not to activate naïve cells. We additionally find that ACPA+ RA patient T cells specific for these peptides are frequently central and effector memory T cells, but not as frequently Tregs, suggesting that they have, in fact, been activated in the periphery and are not tolerized in RA patients. However, you are correct that we cannot determine whether these cells are directly pathogenic *in vivo*, even if they have been activated, and we have added notes to the discussion to make that limitation clear.

Finally, we have made it much clearer in the results when differences were significant or not and highlighted the associated *P* values in the text, particularly throughout Fig. 6 and 7. We believe this has markedly improved the interpretation and clarity of our results, and we appreciate you bringing these concerns to our attention.

Reviewer #3 acting as mediator for comments from Reviewer 1 (absent; expertise in citrullinated antigens, class-II antigen presentation)

It is correct that autoreactive T cells reactive with unmodified and commonly occurring peptide peptides are frequent in the repertoire. But I guess this is the point, that's the reason these are easily picked up in this assay. The question is which function they have. There is no obvious reason to think they are pathogenic, rather they may have no role (ignorant) or to be regulatory. There is no way to show this in the present work, but the writing should take such considerations into account. The only way to really show functional effects is in corresponding experimental models and data in models of RA shows that there are very specific T cells that actually could mediate arthritis, and these are uniquely identifying posttranslational (glycosylation) modified peptides (on collagen II which is also a citrullination target) and are in fact not completely tolerized in thymus. Such T cell reactivities are conserved between mice and humans and the authors hypothesis is compatible with a role for such T cells. Thus, pathogenic T cells are likely to be exceptions. Having this said, I think with the restrictions the paper has, it will not be possible to make a better experimental work than they have already done, although I think they could improve the discussion maybe including some of the points above.

Thank you for highlighting the both the merits and limitations of our work. The new data we provide in Fig. 7 demonstrates that a large proportion of the autoreactive T cells we identified in RA patients are memory T cells and not Tregs, suggesting that these cells are not ignorant nor regulatory T cells. As shown in Fig. 6d, we can also detect the secretion of effector cytokines in response to stimulation with cryptic peptides. However, we agree that with our experimental setup, we cannot go further towards demonstrating a pathogenic role for these T cells *in vivo*, and we have modified our language in the discussion to reflect this limitation as stated above. We have additionally added a note to clarify that *in vivo* models are necessary to demonstrate a direct pathogenic role for these autoreactive T cells in mediating inflammatory arthritis.